

# 1  Return Levels of Temperature Extremes in Southern Pakistan

Maida Zahid[1], Richard Blender[1], Valerio Lucarini[1,2] and Maria Caterina Bramati[3]
1. Meteorological Institute, University of Hamburg, Hamburg Germany
2. Department of Mathematics and Statistics, University of Reading, Reading, UK
3. Department of Statistical Science, Cornell University, New York, United States
*Correspondence to:* Maida Zahid (maida.zahid@uni-hamburg.de)
**Abstract.** Southern Pakistan (Sindh) is one of the hottest regions in the world and is highly vulnerable to
temperature extremes. In order to improve rural and urban planning, information about the recurrence of
temperature extremes is required. In this work, return levels of the daily maximum temperature $T_{max}$ are
estimated, as well as the daily maximum wet-bulb temperature $TW_{max}$ extremes. The method used is the Peak
Over Threshold (POT) and it represents a novelty among the approaches previously used for similar studies in
this region. Two main datasets are analyzed: temperatures observed in nine meteorological stations in southern
Pakistan from 1980 to 2013, and the ERA Interim data for the nearest corresponding locations. The analysis
provides the 2, 5, 10, 25, 50 and 100-year Return Levels (RLs) of temperature extremes. The 90% quantile is
found to be a suitable threshold for all stations. We find that the RLs of the observed $T_{max}$ are above 50°C in
northern stations, and above 45°C in the southern stations. The RLs of the observed $TW_{max}$ exceed 35°C in the
region, which is considered as a limit of survivability. The RLs estimated from the ERA Interim data are lower
by 3°C to 5°C than the RLs assessed for the nine meteorological stations. A simple bias correction applied to
ERA Interim data improves the RLs remarkably, yet discrepancies are still present. The results have potential
implications for the risk assessment of extreme temperatures in Sindh.
**Key words**
Extreme temperature, return levels, peak over threshold, Generalized Pareto Distribution, declustering.
**1 Introduction**
Extreme maximum temperature events have received much attention in recent years, because of the associated
risk of mortality and their likely increase in intensity and frequency in climate change scenarios (Sheridan and
Allen, 2015). An example of the potential impact of raising maximum temperatures is the recent heat wave in
Southern Pakistan (Sindh), which occurred between June 17th and June 24th 2015 and broke all the records with a
death toll of 1400 people, and over 14000 people hospitalized. The temperatures in different cities of the Sindh
region were in the range of 45°C - 49°C during the event (Imtiaz and Rehman, 2015). Karachi had the highest
number of fatalities (1200 people approximately). The Pakistan Meteorological department issued a technical
report stating a very high heat index (measuring the heat stress on humans due to high temperature and relative
humidity) during this heat wave (Chaudhry et al., 2015).
In summer, Sindh becomes very hot and with the arrival of a monsoon the humidity increase in the region
(Chaudhry and Rasul, 2004). This lethal combination of high temperature and relative humidity is known as wet-
bulb temperature, which increases the death rates, and severely impacts the human habitability (Pal and Eltahir
2015). The human body generally maintains the temperature around 37°C. However, the human skin regulates at



or below 35°C to release heat (Sherwood and Huber, 2010). Under high levels of the moisture content in the
atmosphere, the human body cannot maintain the skin temperature below 35°C and can develop ailments like
hyperthermia, heat strokes and cardiovascular problems. Hyperthermia can occur even in the fittest human
beings, if they are exposed to an environment where wet-bulb temperature is greater than 35°C for at least six
hours. Hyperthermia is a condition where extremely high body temperature is reached, resulting from the
inability of the body to get rid of the excess heat. It occurs mostly when temperature and relative humidity levels
are extremely high at the same time.
This study devotes special attention to Sindh because of its exposure to the frequent and intense temperature
extremes in the past (Zahid and Rasul, 2012). This region is considered as one of the most vulnerable regions in
Pakistan. Sindh stretches from 23.5° N – 28.5° N and 66.5°E - 71.1°E, and is bounded on the west by the Kirthar
Mountains, to the north by the Punjab plains, to the east by the Thar desert and to the south by the Arabian Sea
(Indian Ocean) and in the center fertile land around Indus river. The Indus river is the source of water for the
agriculture lands. Cotton, wheat and sugar cane are grown on the left bank of the Indus and rice, wheat and gram
on the right bank (Chaudhry and Rasul, 2004). Cotton is the cash crop of the country.
The climate in Sindh is arid and subtropical with less than 250 mm annual rainfall. The temperature frequently
exceeds 45°C in summer (May-September) and the minimum average temperature recorded during winter
(December- January) is 2°C. Table 2 shows the mean monthly climatic characteristics of the region from 1980-
2010. Figure 1 shows the spatial distribution of all nine weather stations of Pakistan meteorological department,
and the ERA Interim grid points close to the corresponding locations. High population density, limited resources,
poor infrastructure and high dependence of the local agriculture on climatic factors, mark this region as highly
vulnerable to the impacts of climate change.
The Intergovernmental Panel on Climate Change (IPCC) scenarios estimates for this region an increase in the
surface temperature of the order of 4°C in this region by the end of 2100. This may significantly reduce crop
yields, and cause huge economic losses to the country (Islam et al., 2009; Rasul et al., 2012; IPCC, 2012;
Pachauri et al., 2014). Furthermore, it might increase the risks of heat strokes, cardiac arrest, high fever, diarrhea,
cholera and vector borne diseases. Heat waves became more frequent and intense during 90's in Southern
Pakistan. Zahid and Rasul (2010) reports the significant rise in heat index and heat waves events longer than ten
days in Sindh. The enhanced mortality rate related to the heat waves is a serious problem, and two obvious
examples are the 1991 and the previously mentioned 2015 heat waves (Imtiaz and Rehman, 2015).
The analysis of extreme climatic events is a very active area of research in geosciences (Christidis et al., 2005,
2010; Tebaldi et al., 2006; Zwiers et al., 2011; Morak et al., 2011, 2013). In order to facilitate and standardize the
analysis of extremes, the World Meteorological Organization (WMO) has suggested 27 specific climate indices,
like the number of hot days, cold days, wet days, dry days, etc. (Tank et al., 2006; 2009, Frisch et al., 2002; Choi
et al., 2009; Lustenberger et al., 2014). The investigation and analysis of such climate indices has now reached a
high level of popularity.





Extreme value theory (EVT) represents an increasingly widespread approach in climate studies (Coles, 2001,
Zhang et al., 2004; Brown et al., 2008; Faranda et al., 2011; Acero et al., 2014) to estimate the occurrence of the
extreme events. The peak over threshold (POT) approach determines the distribution of the exceedances above a
threshold. The exceedances are asymptotically distributed according to the Generalized Pareto Distribution
(GPD). GPD has remarkable properties of universality when the asymptotic behavior is considered (Lucarini et
al., 2016), while one can expect that the threshold level above which the asymptotic behavior is achieved depends
on the specifics of the analyzed time series. In particular, when looking at spatial fields, it will depend on the
geographical location.
In this study, we have chosen to use the POT method to assess the temperature extremes in the Sindh region,
because it is the most practical approach in modeling the risks of extremes. It is applied for studying temperature
extremes in different regions of the world (Burgueño et al., 2002; Nogaj et al., 2006; Coelho et al., 2008; Ghill et
al., 2011). However, to our knowledge, the POT method has never been used to analyze the risk of temperature
extremes in Sindh. The POT approach allows in principle for estimating the return periods and the return levels
(RLs) also for time ranges longer than what has been currently observed. This information and this predictive
power can be beneficial for policy makers and other stakeholders. Note that this is exactly the kind of information
planners need when, e.g., designing infrastructures that are deemed to last a very long time.
It is useful to consider two indicators of extreme temperatures: (1) temperature extremes $T_{max}$, and (2) Wet-bulb
temperature extremes $TW_{max}$, and are interlinked, but rarely studied together. The southern Pakistan (Sindh)
lacks the information about both the temperature extremes and faces the consequences of heat waves almost
every year. Thus, considering the need and relevance of the information such a study is necessary and timely.
Therefore, we estimate the return levels of extreme daily maximum temperatures $T_{max}$ and daily maximum wet-
bulb temperatures $TW_{max}$ over the different return periods in Sindh. We apply the peak over threshold (POT)
method on the observational data of the nine weather stations provided by Pakistan meteorological department,
and the ERA Interim data of European center for medium range weather forecast (ECMWF) model for the
corresponding grid points from 1980 to 2013. If the ERA Interim dataset characterizes well the extremes, it could
be an option for the regions inside Sindh where no observational data is available. Furthermore, a standard bias
correction is applied on the ERA Interim data to improve the results.
The paper is organized as follows. In Section 2, the statistical modeling of extremes using peak over threshold
method is briefly illustrated along with a description of the data used. The estimation of daily maximum wet-bulb
temperature is discussed in detail in this Section. Section 3 presents the main results of the POT analysis on the
meteorological station observations, ERA Interim, and bias corrected ERA Interim daily maximum temperature
$T_{max}$ and wet-bulb temperature $TW_{max}$ data at nine locations, viz. Jacobabad, Mohenjo-daro, Rohri, Padidan,
Nawabshah, Hyderabad, Chhor, Karachi, and Badin. The performance of the ERA Interim and bias corrected
ERA Interim in comparison to observations is also described in Section 3. All computations and graphics in this
work are done using the R free open source statistical software, using the packages ismev and extRemes (see
www.R-project.org and R Development core team 2015). Section 4 summarizes the major findings of the study
and concludes our work.





**2. Data and Methodology**
**2.1 Meteorological Station Data**
The daily maximum temperature and relative humidity data recorded at nine meteorological stations in Sindh
from 1980 to 2013 are provided by the Pakistan Meteorological Department (see Table 1).  We select nine
stations, which contain a negligible amount of missing values after 1980, and are suitable for the POT analysis.
An additional criterion is that only those stations are chosen where no changes occurred in measuring instruments
during the last 33 years (Brunetti et al., 2006). None of the station data shows gaps with a duration longer than
two days, which are treated by replacing the missing values with the average of the two previous values.
The temperature data are discretized unevenly with intervals up to 1 degree Celsius. Deidda and Puliga (2006)
uses a Monte Carlo study for simulating various resolutions to show that the discretization in precipitation data
affects the convergence of parameter estimation in the extreme value analysis. For this reason, we produce high
resolution data to compensate the effect of discretization and thus to improve the convergence of the estimator.
To convert station data to higher resolution, we add them to a uniform noise with the magnitude corresponding to
the discretization steps (1 degree C). The noise $r$ is a uniform random variable in the interval [-0.5, 0.5]. The
main property of this noise is to round $(T+r) = T$, where T is the temperature with 1-degree resolution and
'round' is the numerical function, which maps the interval [T-0.5, T+0.5] to T. Thus, adding the noise does not
perturb the information content of the observations. This procedure is applied to all temperature data, irrespective
of the actual resolution, and replicated 100 times using a Monte Carlo approach. Results are then averaged. We
check the influence of this noise parameterization and find no significant bias in the return level estimates.
**2.2 ERA Interim Reanalysis Data**
The gridded daily maximum temperature and relative humidity data of ERA Interim reanalysis is downloaded
from the website  ECMWF Public Datasets web interface (http://apps.ecmwf.int/datasets/). The ERA Interim is
produced from the European center for medium range weather forecast (ECMWF) model with resolution 0.75° ×
0.75° (Dee et al., 2011). The gridded data is then extracted at the closest grid point of all stations, for the period
1980-2013. The latitude and longitude of the ERA Interim stations are displayed in Table 1.

One of the main requirements to perform the POT analysis is a stationary time series.  Therefore, similar to
Bramati et al. (2014), the ADF test of stationarity (Dickey and Fuller, 1979) is performed on all the time series.
The test results show no sign of long-term correlations in the data. High short-term correlations (daily time scale)
typically lead to clusters of extreme values and require the use of a declustering method (see more detail in
Section 2.4).
**2.3 Wet-bulb Temperature Calculations**

The wet-bulb temperature measures the heat stress better than other existing heat indices, because it establishes
the clear thermodynamic limit on heat transfer that cannot be overcome by adaptations like clothing, activity and



acclimatization (Pal and Eltahir 2015, Sherwood and Huber, 2010). Here, we use an empirical equation
developed by Stull (2011) to measure the wet-bulb temperature [°C ].
$$TW_{max} = T_{max} \; atan\left(\alpha_1 \sqrt{RH_{max} + \alpha_2}\right) + atan(T_{max} + RH_{max}) - atan(RH_{max} + \alpha_3) +$$
$$+ \; \alpha_4(RH_{max})^{\frac{3}{2}} \; atan(\alpha_5 RH_{max}) - \alpha_6 \qquad\qquad (1)$$
where $TW_{max}$ is the maximum wet-bulb temperature [°C], $T_{max}$ is the maximum temperature [°C], and $RH_{max}$ is
the maximum relative humidity [%]. This relationship is based on an empirical fit, as in Stull (2011), where the
coefficient values are $\alpha_1$ = 0.151977, $\alpha_2$ = 8.313659, $\alpha_3$ = -1.676331, $\alpha_4$ = 0.00391838, $\alpha_5$ = 0.023101, and
$\alpha_6$ = 4.686035. The Eq. (1) covers a wide range of relative humidity and air temperatures with an accuracy of
0.3°C.
**2.4 Peak Over Threshold**
In order to determine return levels (RLs) of extreme maximum temperatures and maximum wet-bulb
temperatures in Sindh, the Peak Over Threshold approach (POT) is applied to the meteorological stations, the
ERA Interim, and the bias corrected ERA Interim data. In this analysis, extremes are defined as exceedances over
threshold distributed according to the Generalized Pareto Distribution (GPD), which is characterized by two
parameters, the shape $\xi$ and the scale $\sigma$. The GPD for exceedances $x - u$ of a random variable $x$ reads as

$$G(x) = 1 - \left[1 + \xi\left(\frac{x-u}{\sigma}\right)\right]^{-\frac{1}{\xi}} \qquad (x > u, \xi \neq 0 ), \qquad (2)$$

where $u$ is the threshold. The choice of the threshold $u$ is done in order to ensure that the model in (2) provides a
reasonable fit to exceedances of this threshold. The result for the two parameters shape $\xi$ and scale $\sigma$ depend on
the threshold u (Coles, 2001). The shape parameter $\xi$ determines the tail behavior while the scale parameter $\sigma$
measures the variability. For a negative shape parameter, $\xi < 0$, the distribution is bounded (beta distribution), for
vanishing shape parameter, $\xi = 0$, the distribution is exponential, and for a positive shape parameter, $\xi > 0$, the
distribution has no upper bound (Pareto distribution).

In particular, for a negative shape parameters $\xi < 0$ the GPD has an upper bound

$$A_{max} = u - {\sigma}/{\xi} \qquad\qquad (3)$$


$$G(x) = 0 \qquad\qquad (x > A_{max}, \xi < 0 )$$

where $A_{max}$ is an absolute maximum (Lucarini et al., 2014). The choice of the optimal threshold for performing
statistical inference from a time series is crucial. A too large value for $u$ would reduce the number of exceedances
to a few values, inflating the variance of the estimators and by consequence the analysis would unlikely yield any
useful results. On the other hand, a too small value for $u$ would violate the asymptotic nature of the model, with
a possible biased estimation and wrong model selection (Coles, 2001).

The threshold selection is the first step in the application of POT approach, and the stability of the shape





parameters ξ and the scale parameters σ fitting the GPD is assessed with various thresholds. The threshold chosen for each station is the lowest value which  stabilizes the estimates shape parameters ξ and the modified scale parameters σ* (see details later in Section 3.1). The shape ξ, the scale σ and the return levels are estimated using the Maximum Likelihood Estimator (MLE) using the R software (R Development core team 2015), which also provides an standard errors of estimates.

 Multi-occurrence is an important characteristic of extreme climatic events and is referred to as clustering. These clusters are consecutive occurrences of above threshold events. It is important to treat the clustered extremes to achieve the independence assumption, which is crucial for the POT model, in order to apply MLE. We treated the clusters using the concept of Extremal Index (EI) (see Newell, 1964, Loynes, 1965, O'Brien, 1974, Leadbetter, 1983, Smith, 1989, Davison and Smith, 1990). The Extremal Index θ measures the degree of clustering of extremes. It ranges between 0 and 1, (θ = 0 means strong clustering,   θ = 1 absence of clusters). Leadbetter (1983) interprets 1/θ as the mean number of exceedances in a cluster.

The extremal index θ can be estimated in two separate ways. Here, we apply the 'intervals estimator' automatic declustering by Ferro and Segers (2003). A distinctive property of this method is that it avoids the subjective choice of cluster parameters. The main ingredient is an asymptotic result for times between threshold exceedances. The exceedance times are split into two types, a set of vanishing intra-exceedance times within the clusters, and an exponentially distributed set of inter-exceedance times between clusters. The method is iterative starting with largest return times and stops when a limit for the inter-exceedance times is reached. The standard errors of the estimated parameters is obtained by a bootstrap procedure. In this study, the extremal index value is ≤ 0.5 in all the time series referring to the clusters.

The primary focus of the study is to estimate N - years return levels (RLs) $x_N$, which is exceeded on the time scale of N years (Coles, 2001) and reads

$$x_N = u + \frac{\sigma}{\xi}\left[(Nn_y\zeta_u)^\xi - 1\right],\qquad(4)$$

where N represents the return period, $n_y$ is the number of observations per year , $\zeta_u$  is the probability of an individual observation exceeding the threshold $u$, the shape parameter is  ξ and the scale parameter is σ.

**2.5. Bias Correction Method**

A simple bias correction is applied to each ERA Interim time series through a rescaling that adjust the first two moments (mean and variance) to the sample moments calculated on the corresponding observations. Therefore, the bias correction is applied to the entire time series and it is not tailored to the extreme events only. The bias corrected ERA Interim time series $x$ is expressed as



$$x = \bar{z} + \frac{y_{ERA} - \bar{y}}{\sigma_y} \cdot \sigma_z$$

(5)

where $y_{ERA}$ is the ERA Interim time series, $\bar{y}$ and $\sigma_y$ its mean and standard deviation, whereas $\bar{z}$ and $\sigma_z$ are the mean and standard deviation of the meteorological station temperatures. The bias corrected ERA Interim time series shows better results compared to the original ERA Interim data. The comparison of extremes as detected in the station observations, in the ERA Interim, and in the bias corrected ERA Interim time series is carried out in Section 3.

**3. Results and Discussion**

**3.1 Threshold Selection**

The threshold selection is the first step in a POT analysis. It is essential to choose a threshold that is high enough to be in the asymptotic limit of the distribution of exceedances, but low enough to have ample data for the fit. The threshold selection is performed using diagnostic plots of the modified scale parameter $\sigma^*$ ($\sigma^* = \sigma u - \xi u$) and the shape parameter $\xi$ of the observed, ERA Interim, the bias corrected ERA Interim $T_{max}$, and $TW_{max}$ in all stations. In GPD, the excesses above a high threshold have same shape but shifted scale. In order to deal with this problem the modified scale $\sigma^*$ is used, because its estimate remains constant above a sufficiently high threshold guaranteeing that the asymptotic properties are obeyed (Sacrrott and MacDonald, 2012).We observe both the modified scale parameter and the shape parameter $\xi$ stability plots carefully. The threshold u is selected as the lowest value where the two parameters are invariant in order to reach the asymptotic limit (Coles, 2001 and Furrer et al., 2010). Figure 2 shows the parameter stability plots of the station observed $T_{max}$ for Karachi only, as an example to explain the threshold selection procedure. We observe that the 90% quantile is an appropriate threshold for all the station observed, the ERA Interim, the bias corrected ERA Interim $T_{max}$, and $TW_{max}$.

In addition to diagnostic plots of the modified scale parameter $\sigma^*$ and the shape parameter $\xi$, the mean residual life plot is used to select the appropriate threshold for the POT analysis. The mean residual life plot is initiated by Davison and Smith, (1990), according to them lowest value of the threshold should be selected when the threshold based mean excesses are consistent. Hence, the threshold is selected when the plot is approximately linear, like in case of Karachi the station observed $T_{max}$ plot appears to be linear and stable at u = 36, indicating u = 36 as the most suitable threshold for Karachi (Figure 3).

**3.2 GPD Fit**

The goodness of fit is evaluated by means of Quantile-Quantile (Q-Q) plots and hypothesis testing. The Q-Q plot analysis is performed for the stations observed, the ERA Interim, the bias corrected ERA Interim daily $T_{max}$ and $TW_{max}$. The Q-Q plots of the observed $T_{max}$ show that the GPD fits well in most of the stations. However, in a few stations the empirical values show slight deviation from the modeled values like Jacobabad, Mohenjo-daro, Padidan and Chhor. In spite of minor deviations at some stations, still most of the exceedances have a good fit with the model. The Q-Q plots of the observed $TW_{max}$ also show good GPD fits in all stations.



The Q-Q plots of the ERA Interim $T_{max}$ indicates that the GPD fits are not good. The empirical values of the
higher quantiles are deviating from the theoretical quantiles in all stations. However, if the higher quantiles are
neglected, then the stations like Jacobabad, Mohenjo-daro, Rohri, Padidan, Nawabshah, Chhor, and Badin shows
that the exceedances fit very well. Likewise, the Q-Q plots of the ERA Interim $TW_{max}$ do not show good fits with
the GPD model. The Q-Q plots of the bias corrected ERA Interim $T_{max,}$ and $TW_{max}$ show better results than the
ERA Interim. We notice that the $T_{max}$ of the ERA Interim and bias corrected ERA Interim fit better than the
$TW_{max}$ if the higher quantiles are ignored.
In order to assess the goodness-of-fit, we apply the Kolmogorov-Smirnov (K-S) test and Anderson-Darling (A-D)
test to the data of meteorological stations, ERA Interim, bias corrected ERA Interim $T_{max}$ and $TW_{max}$. The p-
values indicate a good performance of the fit procedure. Table 3 displays the results of the K-S and A-D statistics
of the $T_{max}$ and $TW_{max}$ in all the data sets.
**3.3 Parameter Estimates**
Here, we analyze the shape parameter $\xi$ , the scale parameter $\sigma$, and  threshold u for all considered datasets. The
standard errors of the shape $\xi$ and the scale $\sigma$ parameters are estimated using the Maximum Likelihood
Estimation (MLE), and given in Table 4. The spatial distribution of the shape parameter $\xi$ and the scale parameter
$\sigma$ of the GPD in Sindh are shown in Figure 4. The shape parameters $\xi$ are all negative in all datasets at all
stations. This is hardly surprising, as meteorological and physical processes make sure that the temperature
cannot grow locally without control. Figure 4 displays the bias corrected ERA Interim results only. The observed
$T_{max}$ shape parameters $\xi$ are between -0.418 to -0.223, and for $TW_{max}$ within -0.323 to -0.177. The bias corrected
ERA Interim $T_{max}$ shape parameters $\xi$ range from -0.305 to -0.002, and $TW_{max}$ are between -0.18 to -0.01.
The scale parameters $\sigma$ of the observed $T_{max}$ are from 2.08 to 2.76, and the $TW_{max}$ are in a range 1.86 to 2.76. In
the ERA Interim analysis, the scale parameter $\sigma$ of $T_{max}$ is within 1.00 - 1.95, and for $TW_{max}$ within 0.74 -1.75.
We observe a difference in the scale parameters of both the observed, the ERA Interim $T_{max}$ and $TW_{max}$. We find
that the scale parameters of the bias corrected ERA Interim data are much closer to those estimated for $T_{max}$ and
$TW_{max}$ using the station data. In the bias corrected ERA Interim $T_{max}$ the scale parameters $\sigma$ are between 1.50 -
2.75, while for $TW_{max}$ are within a range 1.40 – 2.40 (Figure 4).
**3.4 Absolute Maxima**

Once the shape $\xi$ , the scale $\sigma$, and the thresholds u are fixed, it is possible to compute the theoretical absolute
maxima using Eq. (3) (Section 2.4). Theoretical absolute maxima can be compared with the observed ones for
each station to better understand the signals of warming in Sindh. The daily maximum temperature $T_{max}$ and the
maximum wet-bulb temperature $TW_{max}$ (station data, the ERA Interim, and the bias corrected ERA Interim) have
negative shape parameter $\xi$ in all stations. This means that according to Eq. (2) in section 2.4, the probability
distribution function (pdf) is bounded by the maximum values. These maximum values are the theoretical upper
limits predicted by the GPD fit. The analysis shows that the observed absolute maxima $T_{max}$ and $TW_{max}$ in all
stations of the three data sets are below the theoretical absolute maximum, as expected (Figure 5). This gives us
confidence on the quality of our fit. The following piece of information can also be derived. Assume that one





observes in the future an extreme event larger than the maximum inferred in the present dataset; this may suggest
some non-stationarity in the most recent portion of the dataset.
**3.5 Return Levels**
The return levels (RLs) are computed considering various return periods (2, 5, 10, 20, 50, 100-year). The return
level plots of the stations observed, the ERA Interim, the bias corrected ERA Interim daily maximum
temperature $T_{max}$ and daily maximum wet–bulb temperature $TW_{max}$ are displayed in Figures 6 and 7. The return
levels follow the north-south gradient of the climatic mean temperatures. The northern parts of the Sindh are
hotter than the southern parts. Therefore, different stations have different potential for maximum temperature
return levels. The stations located in the North are Jacobabad, Mohenjo-daro, Rohri, Padidan, and Nawabshah.
While Hyderabad, Chhor, Karachi, and Badin are sited in the South.
The 2, 5, 10, 20, 50, 100-year RLs estimated in Sindh for station observed $T_{max}$ at time reach over 50°C in
Jacobabad, Mohenjo-daro, Padidan, Nawabshah, and over 45°C in Rohri, Hyderabad, Chhor, Karachi, Badin.
The ERA Interim $T_{max}$ return levels are at least 3°C to 5°C lower in all stations. However, the ERA Interim $T_{max}$
captures the geographical variability of the field, but cannot estimate the correct magnitude of the events.  For
example, in Badin the return level of the station $T_{max}$ is 42°C in a 3-year return period, while the ERA Interim
show the same value of the return level in a 30-year return period (Figure 6).
The RLs of $TW_{max}$ are over the 35°C in all meteorological stations. As for the ERA Interim RLs of $TW_{max}$ are
greater than 30°C for all the stations except Karachi, which has RLs less than 30°C. Here, we see again that the
RLs of the ERA Interim $TW_{max}$ are smaller than the RLs of station $TW_{max}$. For example, in Badin station, the RLs
of the station $TW_{max}$ is 38°C in a 4-years return period whereas, the ERA Interim reaches the same RLs in a 15-
year return period (Figure 7).

It is important to underline that the bias between the station and the ERA Interim data is rather relevant when one
wishes to address the impact of hot climatic extremes to the active crop production in the region. The crops are
very sensitive to temperature variations, and even a rise of one degree Celsius can cause detrimental changes in
the phenological stages of the crops (Hatfield and Preuger, 2015). Every crop has a certain limit to tolerate the
temperature. When temperature exceeds this limit, the crop yield is drastically reduced. In summer, the
temperature and humidity increase to an extent that there are high chances of a rapid pests spread in the crops.
Sindh produces cotton, wheat, rice, mango, banana, and dates, so a correct estimate of temperature extremes is
very important in order to avoid the crops failure and the reproduction of pests. Therefore, we apply the standard
bias correction on the ERA Interim data to check the alterations in the return levels and return periods of $T_{max}$ and
$TW_{max}$.

The bias corrected ERA Interim $T_{max}$ and $TW_{max}$, show improvements in the return levels (RLs), along with a
good correspondence in each station. In a maximum temperature $T_{max}$ analysis the RLs of the bias corrected ERA
Interim overlap the RLs of the station observations in a range 5-100 years, but do not overlap within a range 2-
5years, in the Nawabshah, Hyderabad, Karachi, and Badin. However, the rest of the stations show no overlaps of





the return levels in both the bias corrected ERA Interim and station observations. In a wet-bulb temperature $TW_{max}$ analysis, the RLs of the bias corrected ERA Interim overlap the RLs of the station observations in Mohenjo-daro, Hyderabad, Chhor, and Badin at some intervals. While, no overlapping of the RLs is detected in rest of the stations, while they differ at some intervals (Figures 6 and 7).

The 2, 5, 10, 20, 50, 100-year RLs of $T_{max}$ for the bias corrected ERA Interim data are greater than 50°C in Jacobabad, Mohenjo-daro, Padidan, Nawabshah, and greater than 45°C in Rohri, Hyderabad, Chhor, Karachi, Badin. As for the $TW_{max}$, the 2, 5, 10, 20, 50, 100-year RLs of the bias corrected ERA Interim exceed 35°C in all stations. Figures 6 and 7 show that the ERA Interim time series improves a lot after the bias correction, but the two data sets still have some quantitative differences.

The extremes of daily maximum wet-bulb temperature $TW_{max}$ are estimated as above the human survivability threshold 35°C throughout the region, so the risk of hyperthermia is very high here. The human habitability in such a warm region is already at risk. The most vulnerable people are those who are involve in the everyday outdoor activities like farming, fishing, building construction, athletes, elderly and infants can have heat strokes, dehydration etc. Therefore, an early warning system is necessary in Sindh, to avoid the crop failure, water shortages and casualties due to the heat stress each year.

We also plot the station and bias corrected ERA Interim $T_{max}$, and $TW_{max}$ return levels spatially for the 5, 10, 25 and 50-year return periods (Figures 8 and 9), as a detailed spatial overview of the temperature extremes in Sindh might be of interest to the policy makers.

## 4. Summary and Conclusion

The main objective of this study is the assessment of the return levels of the extreme daily maximum temperatures $T_{max}$ and wet-bulb temperatures $TW_{max}$ in Southern Pakistan (Sindh). In addition, the performance of the ERA Interim $TW_{max}$ is compared to the weather station $TW_{max}$ to assess the ability to estimating temperature extremes in Sindh. Moreover, a standard bias correction is applied to the ERA Interim data to improve its performance in representing temperature extremes.

In summary, the Peak Over Threshold (POT) method is applied to the daily $T_{max}$ and $TW_{max}$ data of nine observatories and to the corresponding nearest ERA Interim temperature data. Standard declustering technique is applied to all time series to achieve the independence assumption of extremes. The 90% quantile is the appropriate threshold choice for the weather stations, the ERA Interim and the bias corrected ERA Interim maximum temperature and wet-bulb temperature. A Generalized Pareto Distribution (GPD) is fit to both $T_{max}$ and $TW_{max}$ for all three datasets. The results show that the shape parameter $\xi$ is negative for all stations. The scale parameter $\sigma$ estimated on weather station temperatures is much closer to the bias corrected ERA Interim estimates than the original ERA Interim data ones. The theoretical absolute maxima of the time series are higher than the observed absolute maxima in all stations. The Q-Q plots are used to assess the GPD fit, which results to



be acceptable for both $T_{max}$ and $TW_{max}$ station data as compared to the ERA Interim data. However, the bias
corrected ERA Interim shows improved GPD fits than ERA Interim.
Return levels (RLs) of $T_{max}$ and $TW_{max}$ are estimated for the 2, 5, 10, 25, 50, 100-year return periods in all
datasets. The RLs of $T_{max}$ estimated using the meteorological station temperatures are greater than 50°C in
Jacobabad, Mohenjo-daro, Padidan, Nawabshah, and greater than 45°C in Rohri, Hyderabad, Chhor, Karachi and
Badin. While the RLs of $TW_{max}$ in station data are larger than 35°C in the entire Sindh, when using ERA Interim
temperatures, they are estimated as greater than 45°C in Northern Sindh and greater than 40°C in southern Sindh.
The differences in the RLs using the two datasets are between 3°C and 5°C for both shorter and longer return
periods due to the minor variations in the shape and scale parameters. Although the ERA-Interim dataset does not
capture well the magnitude of the extremes, but it provides a good representation of their spatial fields.
A simple standard bias correction is applied to the ERA Interim to assess whether the return levels of extremes
are better predicted after the rescaling is applied. The bias corrected ERA Interim $T_{max}$ and $TW_{max}$ gives return
levels closer to the meteorological stations observed ones than the original ERA Interim return levels at all
stations. Although the bias corrected ERA Interim shows a good correspondence with the meteorological station
data, some differences remain.
This paper contains novel and beneficial information regarding the assessment of the temperature extremes ($T_{max}$
and $TW_{max}$) in Sindh, which would help the local administrations to prioritize the regions in terms of adaptations.
This research fills the gaps in the literature providing information on $T_{max}$ and $TW_{max}$ extremes in Sindh, which
would benefit both public and private stakeholders.
**Acknowledgements**
We like to thank Climate KIC, for funding this research. This publication is a part of a Climate KIC project
"Extreme Events in Pakistan: Physical processes and impacts of changing climate", which belongs to the
adaptation services platform of the Climate KIC. Thanks to Pakistan Meteorological Department (PMD) and the
European Center for Medium range Weather Forecast (ECMWF) for providing datasets. The R development
core team (2015) is acknowledged for providing statistics packages. We would like to thank the DFG Cluster of
Excellence CliSAP for partially supporting this research activity.

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

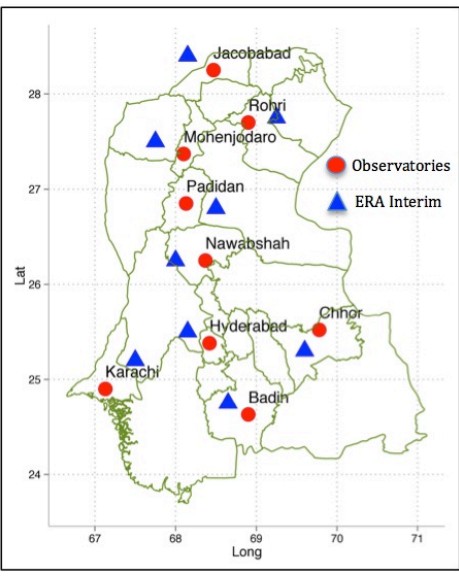

Figure 1: Study Domain (23.5 − 28.5° N , 66.5- 71.1°E)





Table 1. Code, Name, Geographic coordinates and Altitude of the stations.

| Code | Name | PMD weather stations | | | ERA-Interim stations | |
|---|---|---|---|---|---|---|
| | | Latitude | Longitude | Altitude (m) | Latitude | Longitude |
| JCB | Jacobabad | 28° 18'N | 68° 28'E | 55 | 28 °4'N | 68 °15'E |
| MJD | Mohenjo-daro | 27° 22'N | 68° 06'E | 52.1 | 27°5'N | 67 °75'E |
| RHI | Rohri | 27° 40'N | 68° 54'E | 66 | 27°75'N | 69 °25'E |
| PDN | Padidan | 26° 51'N | 68° 08'E | 46 | 26°8'N | 68 °5'E |
| NWB | Nawabshah | 26° 15'N | 68° 22'E | 37 | 26°25'N | 68 °0'E |
| HYD | Hyderabad | 25° 23'N | 68° 25'E | 40 | 25°5'N | 68 °15'E |
| CHR | Chhor | 29° 31'N | 69° 47' E | 5 | 25°3'N | 69 °6'E |
| KHI | Karachi | 24° 54'N | 67°08' E | 21 | 25°2'N | 67 °5'E |
| BDN | Badin | 24° 38'N | 68° 54'E | 10 | 24 °75'N | 68 °65'E |

Table 2. Monthly mean climatic characteristics of all nine stations from1980-2010.

| Stations | Mean Temperature (°C) | | | | | | | | | | | | |
|---|---|---|---|---|---|---|---|---|---|---|---|---|---|
| | Jan | Feb | Mar | Apr | May | Jun | Jul | Aug | Sep | Oct | Nov | Dec | Annual |
| Jacobabad | 15.2 | 18.2 | 24 | 30.5 | 35.6 | 37 | 34.8 | 33 | 31.4 | 27.8 | 22.3 | 16.7 | 27 |
| Mohenjo-daro | 13.9 | 16.7 | 23 | 29.1 | 34.1 | 35 | 33.9 | 32.9 | 30.9 | 26.7 | 21.1 | 15.9 | 25.9 |
| Rohri | 15.6 | 18.2 | 23.6 | 29.8 | 34.5 | 35.6 | 33.9 | 32.3 | 31.2 | 27.6 | 22.1 | 16.9 | 26.4 |
| Padidan | 14.8 | 17.7 | 23.5 | 29.9 | 34.4 | 35.5 | 33.7 | 32.1 | 31 | 27.5 | 22.4 | 16.4 | 26.5 |
| Nawabshah | 15.4 | 18 | 24 | 29.8 | 34.5 | 35.6 | 34 | 32.3 | 31.5 | 28 | 22.4 | 16.9 | 26.7 |
| Hyderabad | 18 | 21 | 26.2 | 30.9 | 33.3 | 34 | 32.4 | 31.1 | 31 | 29.6 | 24.8 | 19.6 | 27.6 |
| Chhor | 16.5 | 19.5 | 25 | 30.1 | 33.5 | 33.7 | 31.6 | 30.1 | 30.1 | 28.2 | 22.6 | 17.9 | 26.3 |
| Karachi | 18.6 | 21.2 | 25.4 | 28.9 | 31.1 | 31.9 | 30.5 | 29.2 | 29.5 | 28.9 | 24.6 | 20.4 | 26.4 |
| Badin | 17.5 | 20.5 | 25.8 | 30.1 | 32.6 | 32.8 | 31 | 29.6 | 29.6 | 28.7 | 24 | 19 | 26.6 |

| Stations | Minimum Temperature (°C) | | | | | | | | | | | | |
|---|---|---|---|---|---|---|---|---|---|---|---|---|---|
| | Jan | Feb | Mar | Apr | May | Jun | Jul | Aug | Sep | Oct | Nov | Dec | Annual |
| Jacobabad | 7.9 | 10.9 | 16.6 | 22.4 | 27.4 | 29.8 | 29.3 | 28.4 | 26.3 | 20.5 | 14.3 | 8.9 | 19.9 |
| Mohenjo-daro | 4.7 | 7.9 | 13.3 | 18.9 | 24 | 27.4 | 27.9 | 27 | 24.7 | 18.2 | 11.8 | 7.3 | 17.3 |
| Rohri | 8.3 | 10.8 | 15.9 | 21.7 | 26.1 | 27.7 | 27.1 | 26 | 24.4 | 19.9 | 14.2 | 9.6 | 18.7 |
| Padidan | 6.5 | 8.9 | 14.5 | 20.2 | 24.7 | 27 | 26.9 | 25.8 | 23.7 | 18.3 | 12.4 | 7.6 | 17.8 |
| Nawabshah | 6.3 | 8.7 | 14.2 | 19.4 | 24.6 | 27.3 | 27.2 | 25.9 | 23.8 | 18.4 | 12.4 | 7.8 | 17.9 |
| Hyderabad | 11.4 | 13.9 | 18.8 | 22.8 | 26.1 | 27.9 | 27.6 | 26.5 | 25.4 | 22.5 | 17.4 | 13 | 21.1 |
| Chhor | 5.9 | 8.9 | 14.8 | 20.3 | 24.8 | 26.9 | 26.5 | 25.3 | 23.9 | 18.7 | 11.8 | 7 | 17.6 |
| Karachi | 11.5 | 14 | 18.6 | 23 | 26.6 | 28.3 | 27.6 | 26.3 | 25.6 | 21.9 | 16.8 | 12.7 | 20.7 |
| Badin | 9.9 | 12.6 | 17.9 | 22.3 | 25.7 | 27.6 | 27.1 | 26 | 25 | 22.1 | 16.5 | 11.4 | 20.2 |

| Stations | Maximum Temperature (°C) | | | | | | | | | | | | |
|---|---|---|---|---|---|---|---|---|---|---|---|---|---|
| | Jan | Feb | Mar | Apr | May | Jun | Jul | Aug | Sep | Oct | Nov | Dec | Annual |
| Jacobabad | 22.6 | 25.6 | 31.4 | 38.6 | 43.9 | 44.4 | 40.2 | 37.6 | 36.8 | 35.1 | 30.3 | 24.4 | 34.1 |
| Mohenjo-daro | 23.1 | 26.2 | 32.1 | 38.7 | 43.8 | 44.2 | 40.9 | 38.7 | 37.5 | 35.2 | 30.5 | 24.8 | 34.5 |
| Rohri | 22.6 | 25.6 | 31.2 | 38.1 | 43 | 43.5 | 40.5 | 38.3 | 37.8 | 35.2 | 30 | 24.3 | 34 |
| Padidan | 23.1 | 26.4 | 32.2 | 39.4 | 43.9 | 44.1 | 40.6 | 38.4 | 38.3 | 36.3 | 31.1 | 25.3 | 34.8 |
| Nawabshah | 24.5 | 27.9 | 33.8 | 40.2 | 44.2 | 43.9 | 40.7 | 38.8 | 39 | 37.7 | 32.3 | 26.1 | 35.5 |
| Hyderabad | 24.7 | 28.1 | 33.7 | 38.8 | 41.3 | 40 | 37.2 | 35.6 | 36.3 | 36.7 | 31.9 | 26.2 | 34.1 |
| Chhor | 26.9 | 29.9 | 35.2 | 40 | 42 | 40.6 | 36.8 | 34.9 | 36.3 | 37.6 | 33.5 | 28.7 | 35 |
| Karachi | 26.3 | 28.4 | 32.2 | 34.7 | 35.5 | 35.4 | 33.3 | 32.1 | 33.2 | 35.5 | 32.5 | 28.2 | 32 |
| Badin | 25.2 | 28.3 | 33.7 | 37.8 | 39.4 | 37.9 | 34.9 | 33.2 | 34.2 | 35.2 | 31.4 | 26.5 | 32.9 |



Table 3. Results of the Kolmogorov-Smirnov Goodness of fit test and Anderson-Darling test between
3         empirical and GPD fits.

| **Observed T*max*** | | | | | | | | | | |
|---|---|---|---|---|---|---|---|---|---|---|
| **Test Statistics** | **Null Hypothesis** | **P-value** | | | | | | | | |
| | | **JAC** | **MJD** | **RHI** | **PDN** | **NWS** | **HYD** | **CHR** | **KHI** | **BDN** |
| Kolmogorov Smirnov | Equality of probability distribution | 0.947 | 0.340 | 0.996 | 0.139 | 0.941 | 0.385 | 0.928 | 0.306 | 0.666 |
| Anderson Darling | Equality of probability distribution | 0.553 | 0.978 | 0.654 | 0.857 | 0.157 | 0.649 | 0.233 | 0.869 | 0.145 |
| **ERA Interim T*max*** | | | | | | | | | | |
| **Test Statistics** | **Null Hypothesis** | **P-value** | | | | | | | | |
| | | **JAC** | **MJD** | **RHI** | **PDN** | **NWS** | **HYD** | **CHR** | **KHI** | **BDN** |
| Kolmogorov Smirnov | Equality of probability distribution | 0.169 | 0.125 | 0.553 | 0.456 | 0.322 | 0.187 | 0.419 | 0.456 | 0.332 |
| Anderson Darling | Equality of probability distribution | 0.355 | 0.263 | 0.165 | 0.587 | 0.615 | 0.398 | 0.266 | 0.687 | 0.425 |
| **Bias corrected ERA Interim T*max*** | | | | | | | | | | |
| **Test Statistics** | **Null Hypothesis** | **P-value** | | | | | | | | |
| | | **JAC** | **MJD** | **RHI** | **PDN** | **NWS** | **HYD** | **CHR** | **KHI** | **BDN** |
| Kolmogorov Smirnov | Equality of probability distribution | 0.452 | 0.4729 | 0.197 | 0.489 | 0.269 | 0.137 | 0.158 | 0.243 | 0.312 |
| Anderson Darling | Equality of probability distribution | 0.352 | 0.315 | 0.235 | 0.270 | 0.335 | 0.289 | 0.216 | 0.390 | 0227 |
| **Observed TW*max*** | | | | | | | | | | |
| **Test Statistics** | **Null Hypothesis** | **P-value** | | | | | | | | |
| | | **JAC** | **MJD** | **RHI** | **PDN** | **NWS** | **HYD** | **CHR** | **KHI** | **BDN** |
| Kolmogorov Smirnov | Equality of probability distribution | 0.981 | 0.111 | 0.341 | 0.226 | 0.457 | 0.545 | 0.441 | 0.385 | 0.211 |
| Anderson Darling | Equality of probability distribution | 0.623 | 0.745 | 0.587 | 0.884 | 0.199 | 0.123 | 0.789 | 0.669 | 0.473 |
| **ERA Interim TW*max*** | | | | | | | | | | |
| **Test Statistics** | **Null Hypothesis** | **P-value** | | | | | | | | |
| | | **JAC** | **MJD** | **RHI** | **PDN** | **NWS** | **HYD** | **CHR** | **KHI** | **BDN** |
| Kolmogorov Smirnov | Equality of probability distribution | 0.712 | 0.564 | 0.955 | 0.425 | 0.258 | 0.134 | 0.856 | 0.497 | 0.222 |
| Anderson Darling | Equality of probability distribution | 0.236 | 0.474 | 0.516 | 0.219 | 0.356 | 0.117 | 0.537 | 0.464 | 0.613 |
| **Bias corrected ERA Interim TW*max*** | | | | | | | | | | |
| **Test Statistics** | **Null Hypothesis** | **P-value** | | | | | | | | |
| | | **JAC** | **MJD** | **RHI** | **PDN** | **NWS** | **HYD** | **CHR** | **KHI** | **BDN** |
| Kolmogorov Smirnov | Equality of probability distribution | 0.268 | 0.688 | 0.127 | 0.372 | 0.268 | 0.229 | 0.591 | 0.582 | 0.478 |
| Anderson Darling | Equality of probability distribution | 0.373 | 0.484 | 0.278 | 0.432 | 0.306 | 0.283 | 0.365 | 0.445 | 0.483 |




Table 4. Estimated parameters shape $\xi$, scale $\sigma$ and standard error $\Delta\xi$ of all the data sets.

| Station observed $T_{max}$ | | | | | | | | | |
|---|---|---|---|---|---|---|---|---|---|
| Estimates | JCB | MJD | RHI | PDN | NWB | HYD | CHR | KHI | BDN |
| Shape $\xi$ | -0.3875 | -0.2550 | -0.4182 | -0.3261 | -0.3323 | -0.3292 | -0.3108 | -0.2225 | -0.3292 |
| Standard Error $\Delta\xi$ | 0.0317 | 0.0226 | 0.0226 | 0.0218 | 0.0208 | 0.0312 | 0.0371 | 0.0341 | 0.0312 |
| Scale $\sigma$ | 2.7540 | 2.0819 | 2.3510 | 2.2144 | 2.1391 | 2.2286 | 2.5629 | 2.5685 | 2.2286 |
| Standard Error $\Delta\sigma$ | 0.1421 | 0.1040 | 0.1075 | 0.1076 | 0.1031 | 0.1166 | 0.1462 | 0.1444 | 0.1166 |
| **ERA Interim $T_{max}$** | | | | | | | | | |
| Estimates | JCB | MJD | RHI | PDN | NWB | HYD | CHR | KHI | BDN |
| Shape $\xi$ | -0.1959 | -0.1788 | -0.2076 | -0.2185 | -0.2135 | -0.3380 | -0.2850 | -0.0376 | -0.2514 |
| Standard Error $\Delta\xi$ | 0.0320 | 0.0348 | 0.0343 | 0.0287 | 0.0265 | 0.0316 | 0.0337 | 0.0508 | 0.0371 |
| Scale $\sigma$ | 1.4643 | 1.3230 | 1.3440 | 1.5045 | 1.5630 | 2.0656 | 1.8497 | 1.3303 | 2.0410 |
| Standard Error $\Delta\sigma$ | 0.0798 | 0.0739 | 0.0741 | 0.0788 | 0.0788 | 0.1082 | 0.0949 | 0.0908 | 0.1153 |
| **Bias Corrected ERA Interim $T_{max}$** | | | | | | | | | |
| Estimates | JCB | MJD | RHI | PDN | NWB | HYD | CHR | KHI | BDN |
| Shape $\xi$ | -0.1959 | -0.1788 | -0.2076 | -0.2185 | -0.2135 | -0.3380 | -0.2850 | -0.0376 | -0.2514 |
| Standard Error $\Delta\xi$ | 0.0320 | 0.0348 | 0.0343 | 0.0287 | 0.0265 | 0.0316 | 0.0337 | 0.0508 | 0.0371 |
| Scale $\sigma$ | 1.9834 | 1.7918 | 1.8205 | 2.0382 | 2.1164 | 2.7980 | 2.3081 | 1.8016 | 2.7636 |
| Standard Error $\Delta\sigma$ | 0.1081 | 0.1001 | 0.1004 | 0.1068 | 0.1068 | 0.1467 | 0.1233 | 0.1229 | 0.1562 |
| **Station observed $TW_{max}$** | | | | | | | | | |
| Estimates | JCB | MJD | RHI | PDN | NWB | HYD | CHR | KHI | BDN |
| Shape $\xi$ | -0.1769 | -0.1860 | -0.2150 | -0.2157 | -0.2164 | -0.3231 | -0.2423 | -0.2190 | -0.1867 |
| Standard Error $\Delta\xi$ | 0.0383 | 0.0354 | 0.0347 | 0.0442 | 0.0266 | 0.0269 | 0.0347 | 0.0368 | 0.0322 |
| Scale $\sigma$ | 2.7590 | 2.0454 | 1.9600 | 2.0780 | 1.8572 | 2.3724 | 2.5126 | 2.3375 | 1.9032 |
| Standard Error $\Delta\sigma$ | 0.1596 | 0.1146 | 0.1084 | 0.1289 | 0.0938 | 0.1191 | 0.1380 | 0.1328 | 0.1055 |
| **ERA Interim $TW_{max}$** | | | | | | | | | |
| Estimates | JCB | MJD | RHI | PDN | NWB | HYD | CHR | KHI | BDN |
| Shape $\xi$ | -0.0896 | -0.0946 | -0.0687 | -0.1257 | -0.1583 | -0.1771 | -0.0902 | -0.0194 | -0.1733 |
| Standard Error $\Delta\xi$ | 0.0379 | 0.0293 | 0.0327 | 0.0342 | 0.0313 | 0.0377 | 0.0357 | 0.0359 | 0.0378 |
| Scale $\sigma$ | 1.2879 | 1.2437 | 1.2311 | 1.4408 | 1.6104 | 1.6499 | 1.3423 | 0.6801 | 1.7886 |
| Standard Error $\Delta\sigma$ | 0.0748 | 0.0660 | 0.0676 | 0.0804 | 0.0875 | 0.0959 | 0.0760 | 0.0398 | 0.1028 |
| **Bias Corrected ERA Interim $TW_{max}$** | | | | | | | | | |
| Estimates | JCB | MJD | RHI | PDN | NWB | HYD | CHR | KHI | BDN |
| Shape $\xi$ | -0.08961 | -0.0946 | -0.06870 | -0.12570 | -0.15831 | -0.17711 | -0.09017 | -0.01942 | -0.17332 |
| Standard Error $\Delta\xi$ | 0.03786 | 0.02931 | 0.03275 | 0.03424 | 0.03134 | 0.03767 | 0.03571 | 0.03593 | 0.03782 |
| Scale $\sigma$ | 1.35674 | 1.64650 | 1.75852 | 1.49477 | 1.52013 | 2.05281 | 2.14609 | 1.39943 | 2.15299 |
| Standard Error $\Delta\sigma$ | 0.07878 | 0.08736 | 0.09651 | 0.08347 | 0.08254 | 0.11924 | 0.12145 | 0.08193 | 0.12370 |




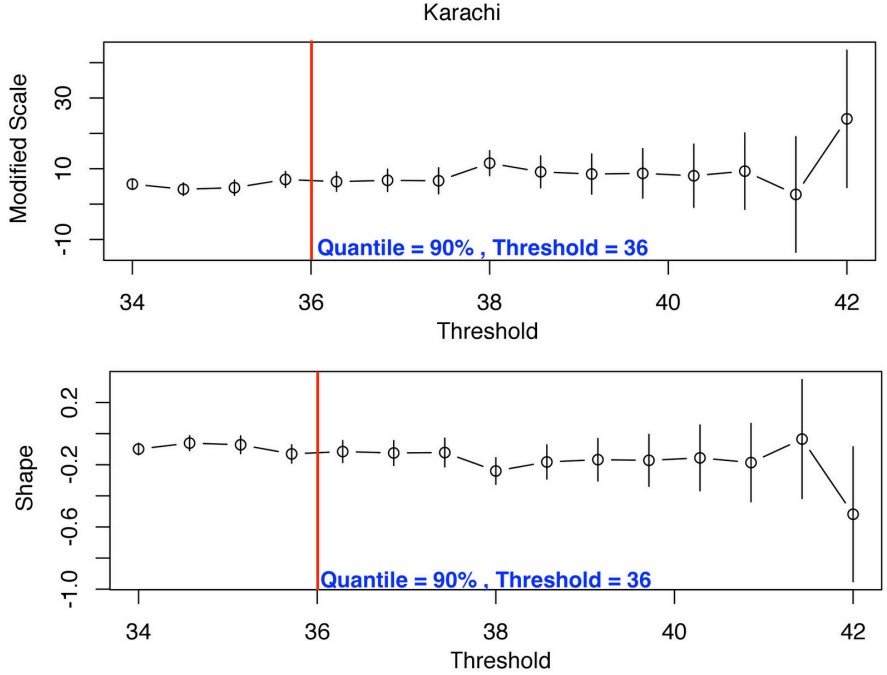

Figure 2. Modified scale ($\sigma^*$) and shape parameter ($\xi$) of the observed $T_{max}$ Karachi. The red vertical lines represent the selected threshold according to the station quantiles


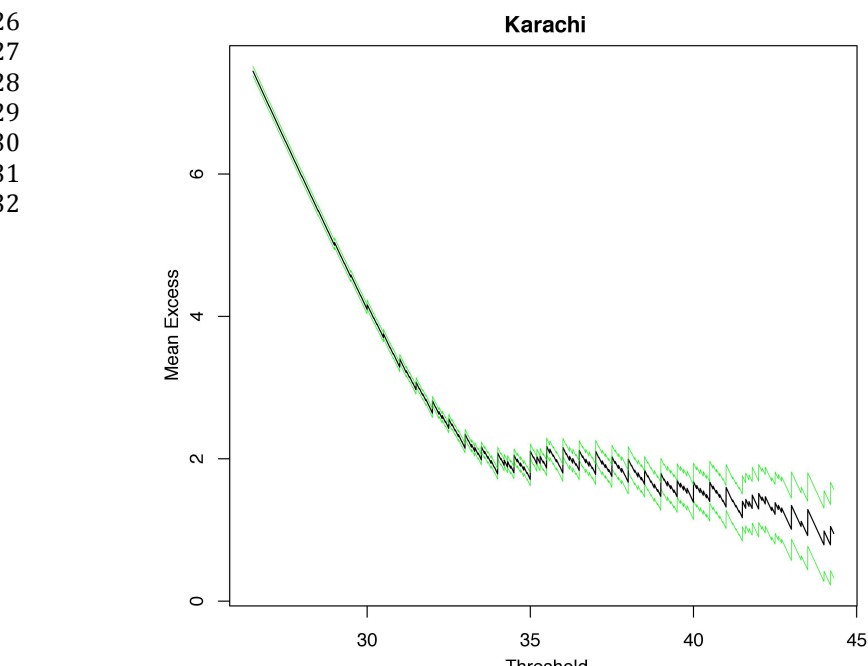

Figure 3. Mean residual life plot of the station observed $T_{max}$ Karachi.



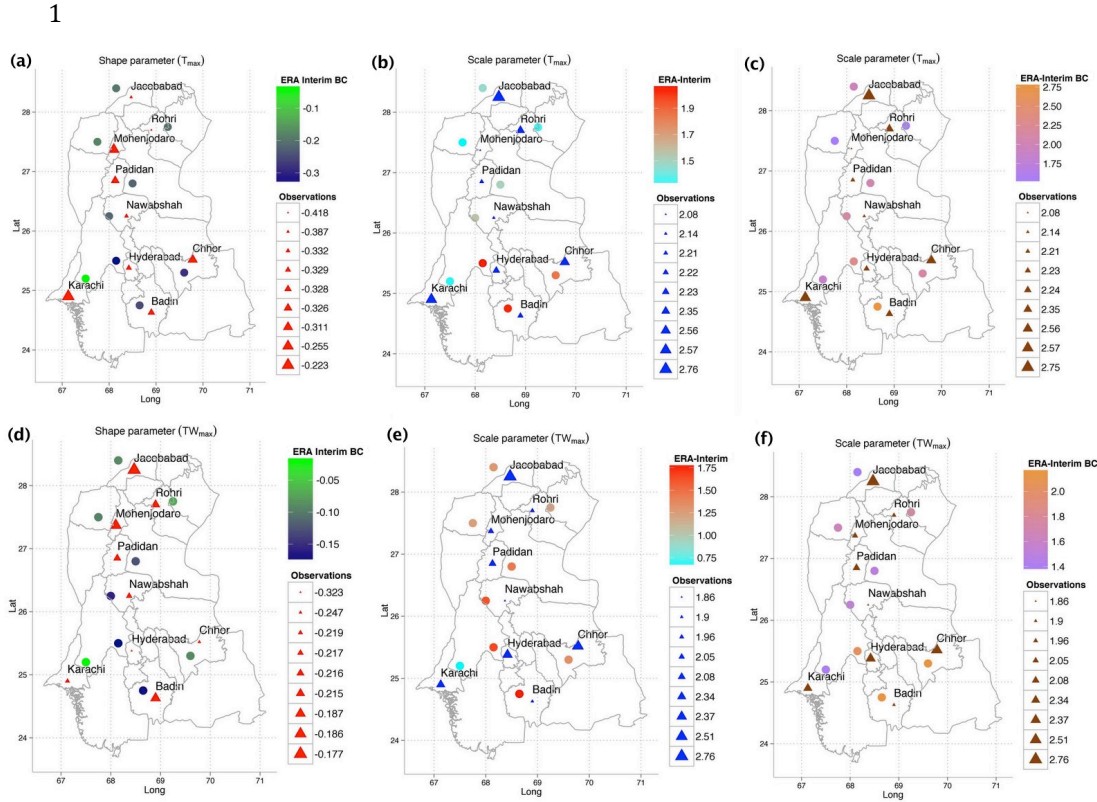

Figure 4. Spatial distribution of the shape parameters ξ and scale parameters σ of the station observed, ERA Interim, and bias corrected ERA Interim T$_{max}$ (upper panel) and TW$_{max}$ (lower panel).





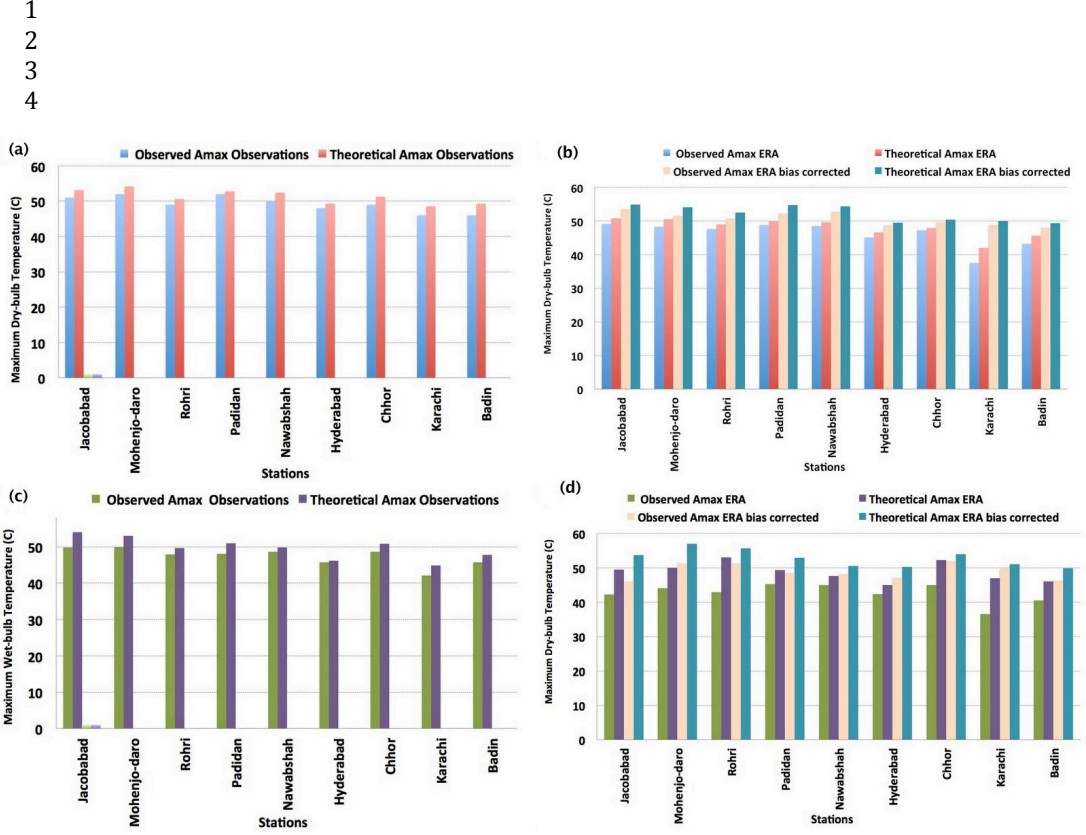

Figure 5. Absolute maxima $A_{max}$ (a) station observed $T_{max}$ (b) ERA Interim and bias corrected ERA Interim $T_{max}$
(c) station observed $TW_{max}$ (d) ERA Interim and bias corrected ERA Interim $TW_{max}$






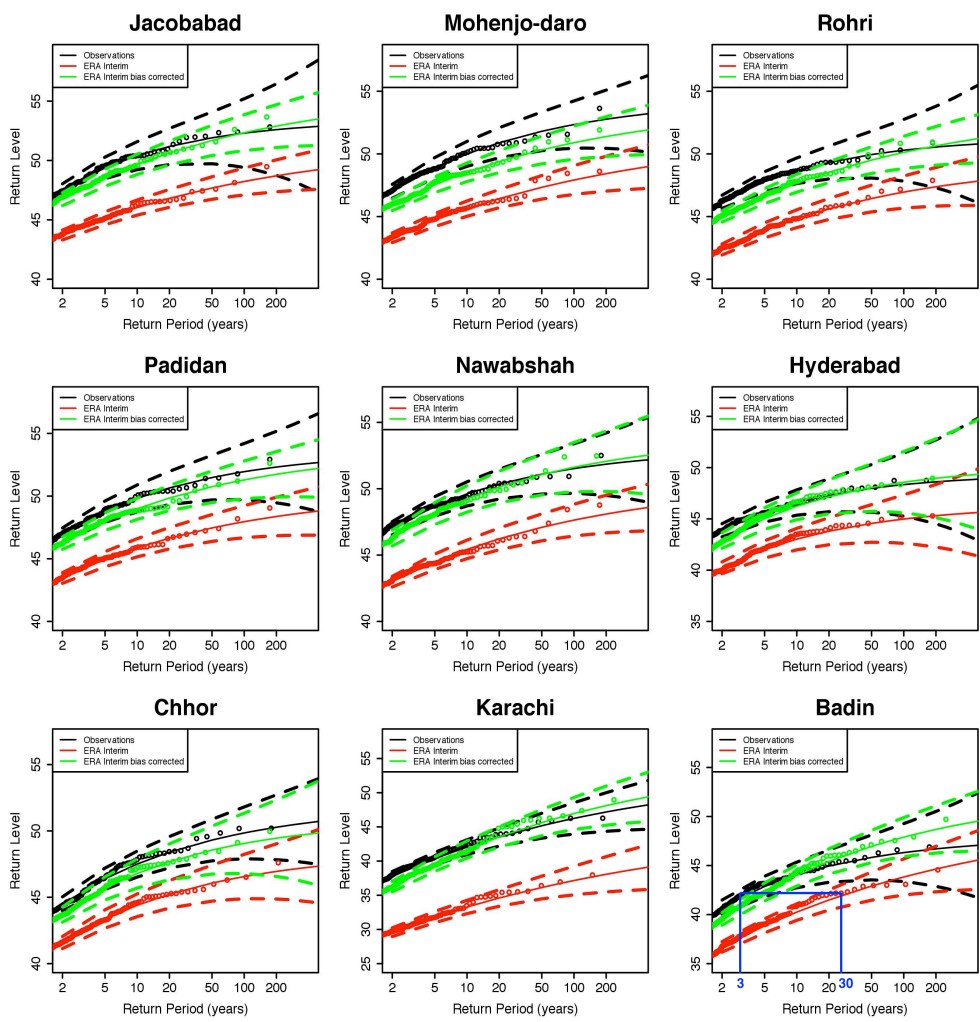

Figure 6. Return level plots of the station observed $T_{max}$ (black) , ERA Interim $T_{max}$ (red), and bias corrected ERA Interim $T_{max}$ (green) in degree Celsius. The blue line is to show a difference in the observed and ERA Interim RLs.




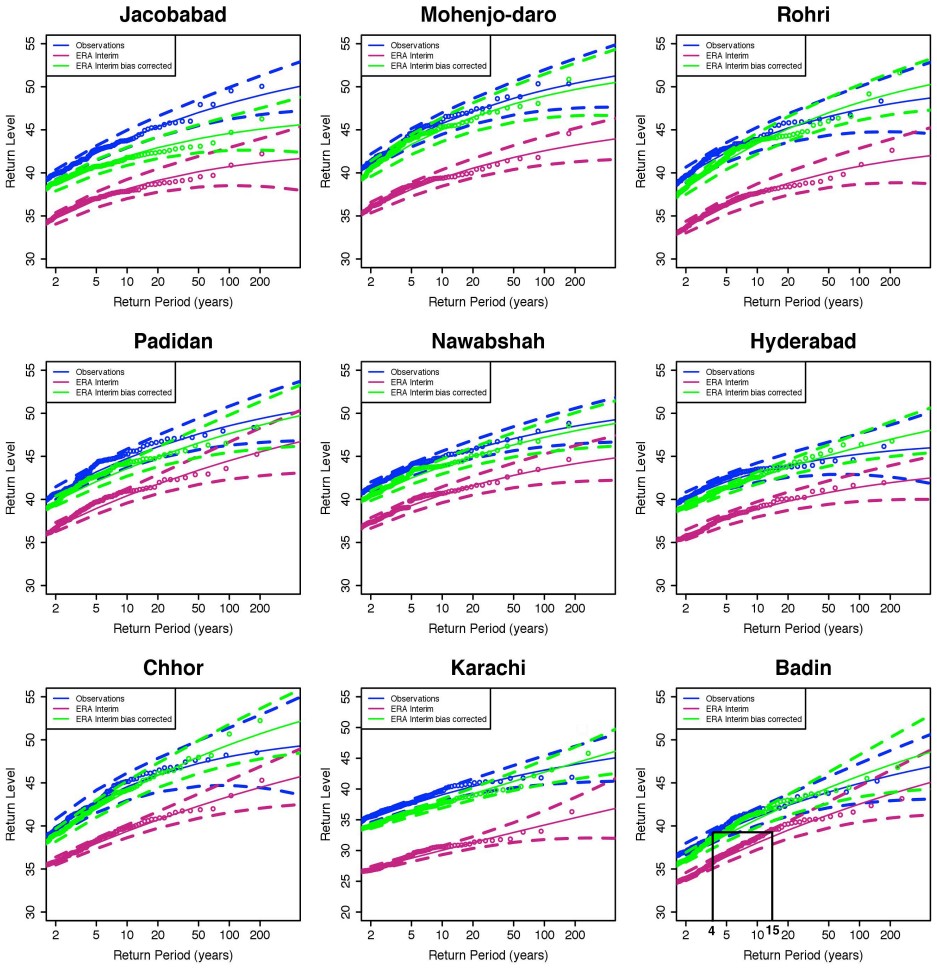

Figure 7. Return level plots of the station observed TW$_{max}$ (blue), ERA Interim T$_{max}$ (pink), and bias corrected ERA Interim T$_{max}$ (green) in degree Celsius. The black line is to show a difference in the observed and ERA Interim RLs.






2
3
4

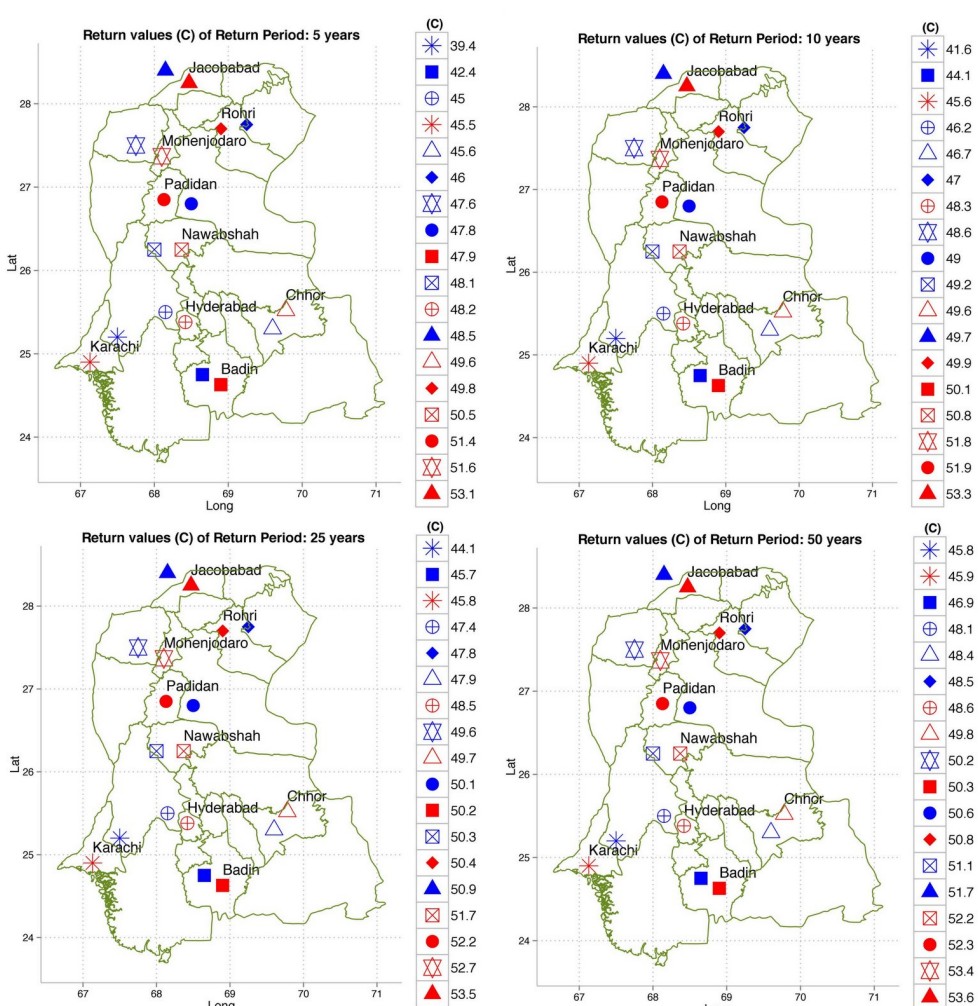

Figure 8.  Spatial distribution of the station observed T$_{max}$ (red) and bias corrected ERA Interim
T$_{max}$ (blue) return levels corresponding to return periods of 5, 10, 25 and 50 years in
southern Pakistan.





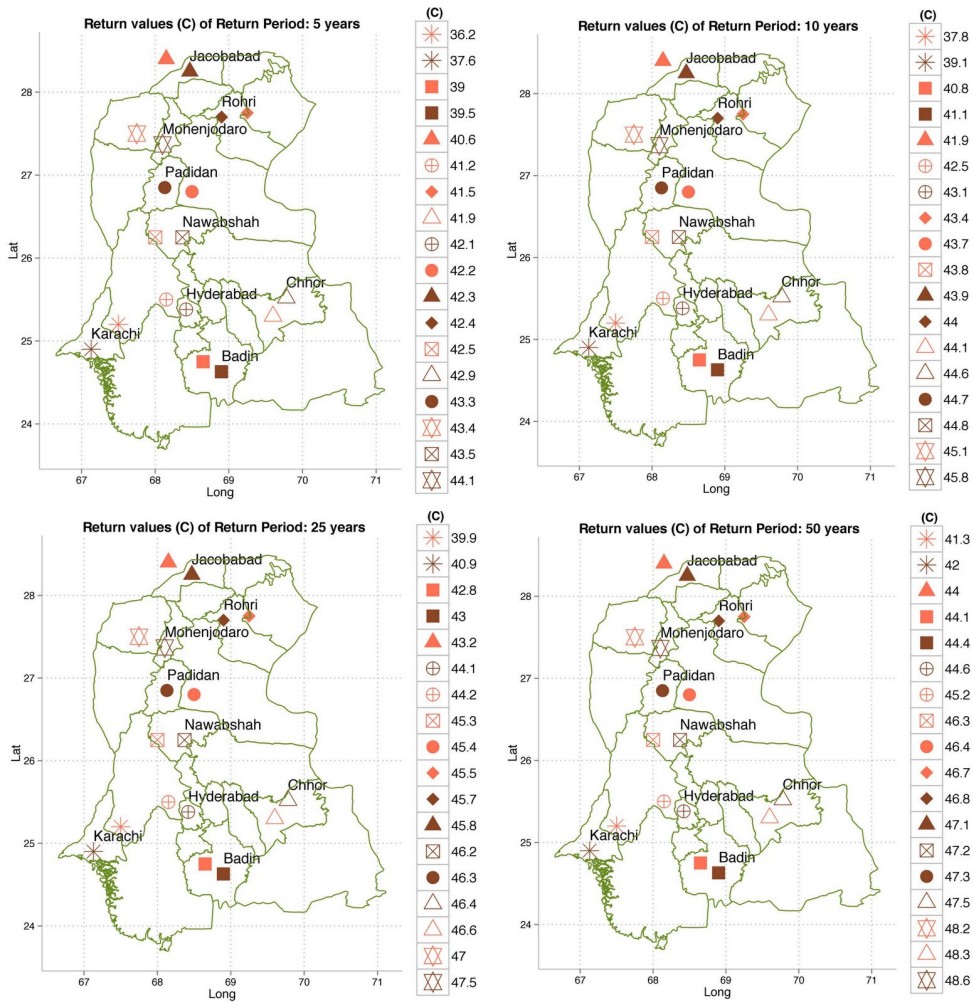

Figure 9. Spatial distribution of the station observed TW$_{max}$ (brown) and bias corrected ERA Interim TW$_{max}$ (orange) return levels corresponding to return periods of 5, 10, 25 and 50 years in southern Pakistan.

