# Peer review of "Return Levels of Temperature Extremes in Southern Pakistan"

_Earth System Dynamics, 2016_

## Referee Comment (RC1) · Anonymous Referee #1 · 18 Jan 2017

This study estimates the return levels and return periods of extreme temperature and wet-bulb temperature in Southern Pakistan using the peak over threshold (POT) approach. Datasets including 33-year daily maximum observations, ERA Interim data, and the bias-corrected ERA Interim data were examined to assess the severity and the likelihood of extreme temperature. This research topic is worthy of investigation, and the results are interesting. I recommend acceptance pending on Major revisions to address the following comments:

1. Both Generalized Extreme Value (GEV) and GPD distributions can be applied for assessing return levels and return periods of climate extreme events. The length of 33-year annual maximum values seems to be sufficient for deriving reasonable estimates using the GEV technique. Therefore, it is not so clear why the GEV distribution is not preferred here. Did authors examine the GEV-based return levels? Does GPD provide

a better fit and a more reliable estimation?

2. GPD approach has been widely applied for estimating the statistics of extreme rainfall and temperature (e.g. Katz et al., 2002, 2010, Cooley et al., 2005, 2007, etc.), and the techniques used in this manuscript are not nascent. In order to sufficiently support the argument that the application of GPD herein is "novel" and can provide "novel" information, highlighting any new or additional findings which can only rely upon the GPD approach is desired.

3. By comparing the distribution parameters and return levels derived from observations and ERA Interim data, it seems that they have a large agreement in the shape parameter estimations at some stations, while the bias in the mean and variance of model simulations is the primary factor that leads to the underestimation of the return levels. Does the agreement in the shape parameters indicate that the underlying physical process which produces extreme temperature is well represented by the climate model, though there is a bias in simulating the internal variability of extreme temperature? Please consider to extend the current discussions in this regard.

4. At stations such as JCB, MJD, RHI, the bias-corrected return levels underestimate the observed values. Which factor/parameter would be responsible for the consistent underestimation? For those locations, is there a way to conduct the bias correction for the shape value? Would a higher threshold correct such underestimation?

5. In section 2.1, the purpose and benefit of adding noise to the data are not clear. By adding the noise, does the convergence of parameter estimation become more efficient? Why?

6. Please consider to rearrange the order of tables and introduce them in sequence. For instance, Table 2 is introduced ahead of Table 1 in the context, so please switch their orders.

7. Please provide the Q-Q plots for the 9 stations, since the authors discussed the

"slight deviation" revealed by examining the corresponding Q-Q plots.

---

## Referee Comment (RC2) · S. Parey (Referee) · 25 Jan 2017

The paper presents estimations of temperature return levels over Southern Pakistan using the Peak Over Threshold approach based on both observation time series and reanalysis time series of the nearest grid points to the observations stations. Such a study is not new research, but as the authors mention, it had not been made for this region, so this justifies publication.

General comments This analysis presents different problems and misunderstandings, therefore I suggest publication after major revisions.

Major comments

1. Line 28 p3, it is stated "If the ERA Interim dataset characterizes well the extremes…"

[Figure]

This is very unlikely, since re-analysis has a too low spatial resolution to represent adequately local extremes, this is not the aim of reanalysis

2. Lines 31 to 35 p4: it is true that stationarity is a requirement to perform POT, but stationarity means that the distribution is invariant by translation in time, not the autocorrelation is weak. Autocorrelation is rather linked to independence, and must be handled too. Maybe here stationarity refers to the iid (independent and identically distributed) condition, then this section deals with independence, but not with identical distribution. However, identical distribution needs to be checked as well, just because of seasonality. Has seasonality been checked? Is the occurrence of the highest values restricted to the summer season or identically distributed throughout the year? If a season appears as favoring high temperature values, then the analysis should be restricted to this season, otherwise the occurrences are not distributed according to a homogenous Poisson process and the frequency of occurrence is biased. Not only seasonality can disturb the identical distribution, trends can too. There is no discussion about possible trends. Maybe it is possible to neglect the trends over the relatively short 1980-2013 period, but this could be checked.

3. When computing TWmax, are we sure that Tmax and RHmax occur in the same time?

4. Some considerations on independence are given again between lines 15 and 22 p6, but no indications are given on how this is used and applied in the study.

5. In section 3.2 concerning the GPD fits, one can read "if the higher quantiles are neglected, then the stations like ... show that the exceedances fit very well". But in an extreme value analysis, the higher quantiles are the targeted ones!

6. In the conclusion, it is stated that: "This paper contains novel and beneficial information ..., which would help the local administration to prioritize the regions in terms of adaptation". What does adaptation mean here? The estimated levels are based on observations, thus these are rare levels which could occur even if there were no

additional warming. It is not clear in the paper if this distinction is made. The notion of return level is defined for stationary time series, that is without any cycles nor trends, and is devoted to the estimation of very rare levels which could happen (once every N years in average), but may not yet have been observed. Climate warming brings other difficulties in their estimation: the definition of a return level has to be changed because a past and a future period are not prone to experience the same temperature levels, and the estimation is complicated. Different papers are devoted to this problem, for example:

Cheng, L., AghaKouchak, A., Gilleland, E., & Katz, R. W. (2014). Non-stationary extreme value analysis in a changing climate. Climatic change, 127(2), 353-369 Du, T., Xiong, L., Xu, C. Y., Gippel, C. J., Guo, S., & Liu, P. (2015). Return period and risk analysis of nonstationary low-flow series under climate change. Journal of Hydrology, 527, 234-250 Obeysekera, J., & Salas, J. D. (2016). Frequency of Recurrent Extremes under Nonstationarity. Journal of Hydrologic Engineering, 21(5), 04016005 Parey, S., Malek, F., Laurent, C., Dacunha-Castelle, D. (2007). Trends and climate evolutions: statistical approach for very high temperatures in France. Clim.Change, 81, 331–352. Parey, S., Hoang, R.T.H., Dacunha-Castelle, D. (2010). Different ways to compute temperature return levels in the climate change context. Environmetrics, 21, 698–718.

---

## Author Comment (AC1) · 13 Feb 2017

The authors would like to thank anonymous refree #1 for the constructive comments that helped us to improve our manuscript. The answers to the comments are given point by point in the following.

1. Both Generalized Extreme Value (GEV) and GPD distributions can be applied for assessing return levels and return periods of climate extreme events. The length of 33-year annual maximum values seems to be sufficient for deriving reasonable estimates using the GEV technique. Therefore, it is not so clear why the GEV distribution is not preferred here. Did authors examine the GEV-based return levels? Does GPD provide a better fit and a more reliable estimation?

Ans: We preferred GPD because in many applications, the POT approach is preferred

to the BM approach for fitting time series because it provides more efficient use of data and has better properties of convergence when finite datasets are considered (Lucarini et al.,2016 ; Coles 2001 ; Holmes and Moriarity; Davison and Smith, 1990). Additionally, we are here interested in investigating the actual tails of the distributions, so the GPD point of view is more appropriate.

2. GPD approach has been widely applied for estimating the statistics of extreme rainfall and temperature (e.g. Katz et al., 2002, 2010, Cooley et al., 2007, Furrer et al., 2010 etc.), and the techniques used in this manuscript are not nascent. In order to sufficiently support the argument that the application of GPD herein is "novel" and can provide "novel" information, highlighting any new or additional findings which can only rely upon the GPD approach is desired.

Ans: We definitely agree that the GPD approach has been applied widely for estimating the statistics of extreme rainfall and temperature. But by saying "novel" we meant that the GPD is applied for the first time to southern Pakistan region, extremes are never been estimated using the GPD here. This paper is the first one to introduce the climate extremes information in the region based on GPD. The word novel has been removed from the manuscript to avoid the misunderstanding.

3. By comparing the distribution parameters and return levels derived from observations and ERA Interim data, it seems that they have a large agreement in the shape parameter estimations at some stations, while the bias in the mean and variance of model simulations is the primary factor that leads to the underestimation of the return levels. Does the agreement in the shape parameters indicate that the underlying physical process which produces extreme temperature is well represented by the climate model, though there is a bias in simulating the internal variability of extreme temperature? Please consider to extend the current discussions in this regard.

Ans: The results for the shape parameter indicate that the functional dependency of the extreme value distribution is reasonably well simulated by the ERA data. This is

especially relevant for the existence of absolute maxima (Eq 3, Section 2.4) in the case of negative shape parameters. The agreement of the shape parameters in the observations and simulations means that the ERA dataset captures an important aspect of extremal behavior. This is in principle a non trivial result, as reanalysis are constructed in such a way that typical conditions are well reproduced.

4. At stations such as JCB, MJD, RHI, the bias-corrected return levels underestimate the observed values. Which factor/parameter would be responsible for the consistent underestimation? For those locations, is there a way to conduct the bias correction for the shape value? Would a higher threshold correct such underestimation?

Ans: The disagreement of the bias corrected results indicates that the standard bias correction method based on the first two moments is not sufficient for these stations. A better agreement could be obtained by including the higher moments to improve the estimate of the extreme values. We do not think that the higher threshold can correct such underestimation because we are within the asymptotic regime.

5. In section 2.1, the purpose and benefit of adding noise to the data are not clear. By adding the noise, does the convergence of parameter estimation become more efficient? Why?

Ans: The advantage of adding a noise is to avoid the spurious statistical effects associated to the presence discrete values assigned to the temperature readings. This is discussed in detail in the cited paper of Deidda and Puliga 2006 for hydrological extremes. Using the described bootstrap method we reduce such problem without biasing the data.

6. Please consider to rearrange the order of tables and introduce them in sequence. For instance, Table 2 is introduced ahead of Table 1 in the context, so please switch their orders.

Ans: The order of tables has been rearranged.
7. Please provide the Q-Q plots for the 9 stations, since the authors discussed the "slight deviation" revealed by examining the corresponding Q-Q plots.

Ans: Q-Q plots are provided as additional material.

References:

Coles, S.: An Introduction to Statistical Modeling of Extreme Values, Springer London, London., 2001.

Cooley, D.; Nychka, D. & Naveau, P. Bayesian spatial modeling of extreme precipitation return levels, J. Am. Statist. Assoc., 102, 824–840.2007

Davison A. C. and Smith R. L : Journal of the Royal Statistical Society. Series B (Methodological) Vol. 52, No. 3, 393-442,1990.

Furrer, E., Katz, R., Walter, M. and Furrer, R.: Statistical modeling of hot spells and heat waves, Clim. Res.,43(3), 191–205, doi:10.3354/cr00924, 2010.

Deidda, R. and Puliga, M.: Sensitivity of goodness-of-fit statistics to rainfall data rounding off, Phys. Chem. Earth, doi:10.1016/j.pce.2006.04.041, 2006.

Holmes, J. D. and Moriarty, W. W.: Application of the Generalised Pareto Distribution to wind engineering, J. Wind Engineering and Industrial Aerodynamics 83, 1– 10.1999

Katz, R. W., M. B. Parlange, and P. Naveau 2002, Statistics of extremes in hydrology, Adv. Water Resour., 25, 1287–1304.2002.

Katz R : Statistics of extremes in climate change. Clim Chang 100(1):71–76.2010.

Lucarini, V., Faranda, D., .Freitas, A.C.M., Freitas, J.M., Holland, M., Kuna, T., Nicol, M., Todd, M., Vaienti, 26 S.: Extremes and Recurrence in Dynamical Systems, John Wiley & Sons Inc,2016.

List of Figures:

Fig1. Quantile-Quantile plots of observed Tmax , u= 90% for 9 stations of Southern

Pakistan (Sindh).

Fig 2. Quantile-Quantile plots of ERA Interim Tmax , u= 90% for 9 stations of Southern Pakistan (Sindh).

Fig 3. Quantile-Quantile plots of Bias corrected ERA Interim Tmax , u= 90% for 9 stations of Southern Pakistan (Sindh).

Fig 4. Quantile-Quantile plots of observed TWmax , u= 90% for 9 stations of Southern Pakistan (Sindh).

Fig 5. Quantile-Quantile plots of ERA Interim TWmax , u= 90% for 9 stations of Southern Pakistan (Sindh).

Fig 6. Quantile-Quantile plots of Bias corrected ERA Interim TWmax , u= 90% for 9 stations of Southern Pakistan (Sindh).

Please also note the supplement to this comment:
http://www.earth-syst-dynam-discuss.net/esd-2016-72/esd-2016-72-AC1-supplement.zip
* * *
**Jacobabad**

**Mohenjo-daro**

**Rohri**

**Padidan**

**Nawabshah**

**Hyderabad**

**Chhor**

**Karachi**

**Badin**

**Fig. 1.**

**Jacobabad**

**Mohenjo-daro**

**Rohri**

**Padidan**

**Nawabshah**

**Hyderabad**

**Chhor**

**Karachi**

**Badin**

**Fig. 2.**

**Jacobabad**

**Mohenjo-daro**

**Rohri**

**Padidan**

**Nawabshah**

**Hyderabad**

**Chhor**

**Karachi**

**Badin**

**Fig. 3.**

![Nine Q-Q plots comparing Empirical Quantiles versus Model Quantiles for the stations Jacobabad, Mohenjo-daro, Rohri, Padidan, Nawabshah, Hyderabad, Chhor, Karachi, and Badin.]

**Fig. 4.**

**Fig. 5.**

[Figure]

Fig. 6.

---

## Author Comment (AC2) · 13 Feb 2017

The authors would like to thank Sylvie Parey (Referee # 2) for the careful review and comments that helped us to improve our manuscript. Our responses to the comments are as follows.

1. Line 28 p3, it is stated "If the ERA Interim dataset characterizes well the extremes: : :".This is very unlikely, since re-analysis has a too low spatial resolution to represent adequately local extremes, this is not the aim of reanalysis

Ans: It is in principle not obvious that ERA data can simulate well meteorological extremes, as reanalysis are constructed in such a way that typical conditions are well reproduced. This is why we look at how well ERA data performs in the target area against observations. Note that Cornes and Jones, (2013) reported that the ERA-

[Figure]

Interim reanalysis data are generally very good at replicating trends in percentile-based measures of temperature extremes. However, ERA-Interim is weak in capturing the extreme temperatures in complex terrains, but our study area has simple terrain.

2. Lines 31 to 35 p4: it is true that stationarity is a requirement to perform POT, but stationarity means that the distribution is invariant by translation in time, not the autocorrelation is weak. Autocorrelation is rather linked to independence, and must be handled too. Maybe here stationarity refers to the iid (independent and identically distributed) condition, then this section deals with independence, but not with identical distribution. However, identical distribution needs to be checked as well, just because of seasonality. Has seasonality been checked? Is the occurrence of the highest values restricted to the summer season or identically distributed throughout the year? If a season appears as favoring high temperature values, then the analysis should be restricted to this season, otherwise the occurrences are not distributed according to a homogenous Poisson process and the frequency of occurrence is biased. Not only seasonality can disturb the identical distribution, trends can too. There is no discussion about possible trends. Maybe it is possible to neglect the trends over the relatively short 1980-2013 period, but this could be checked.

Ans: We agree with the reviewer. We note that one can see non-stationarity and presence of trends as the presence of long-time correlation in the data. Clearly, extreme temperature and extreme heat indices are realized in summer conditions, so the analysis is restricted to summer season. Including the other seasons would make no sense. We have tested that trends are not significant in such a short time interval. Short-time correlations are studied by computing the extremal index $\theta$ in all time series and treated using the associated standard declustering technique.

Additionally, the strict stationarity means that the distribution of the random process is invariant to time shifts. Weak stationarity concerns only the invariance in time of the first and second order moments of the random process, i.e. mean and auto-covariance function (or autocorrelations). However, if the process is Gaussian, then strict stationarity and weak stationarity are equivalent. Nevertheless, weak stationarity and ergodicity are sufficient conditions in time series which allows (when the number of time periods is quite large) to consider the observations 'as if' they were i.i.d. That is why we just check for covariance stationarity (and assume the process is ergodic, i.e. the process has not a long memory: autocovariances decay to 0 for very large time lags). Moreover, when we run the unit-root test (test for stationarity), we check for the presence of a stochastic trend. So, actually, before running the POT analysis it is definitely discussed in the paper the possible presence of a trend.

3. When computing TWmax, are we sure that Tmax and RHmax occur in the same time?

Ans: The occurrence of TWmax takes place in general in different dates than Tmax or RHmax, because one could have a day with high temperature but low humidity and vice versa. Since our study area is next to the Arabian Sea the level of Tmax and RHmax remains constantly high during summer.

4. Some considerations on independence are given again between lines 15 and 22 p6,but no indications are given on how this is used and applied in the study

Ans: Computing the extremal index $\theta$ allows for studying the degree of clustering of extremes. The inverse of $\theta$ gives the average length of a cluster. Usually $\theta = 0$ means strong clustering and dependence, $\theta = 1$ absence of clusters and independence. The extremal index value in all the time series is $\leq 0.5$ referring to dependence. Therefore, it is necessary to decluster the extremes by choosing the largest event in each cluster, before fitting it to the GPD. Note that this is the practical strategy commonly adopted by practitioner as well as the rigorous prescription suggested by mathematics.

5. In section 3.2 concerning the GPD fits, one can read "if the higher quantiles are neglected, then the stations like : : : show that the exceedances fit very well". But in an extreme value analysis, the higher quantiles are the targeted ones!

[Figure]

Ans: By saying neglected we did not mean to neglect the higher quantiles. We wanted to say if the higher quantiles are disregarded or unnoticed. We have replaced the word "neglected", as it can be misleading for the readers.

6. In the conclusion, it is stated that: "This paper contains novel and beneficial information: : :, which would help the local administration to prioritize the regions in terms of adaptation". What does adaptation mean here? The estimated levels are based on observations, thus these are rare levels which could occur even if there were no additional warming. It is not clear in the paper if this distinction is made. The notion of return level is defined for stationary time series, that is without any cycles nor trends, and is devoted to the estimation of very rare levels which could happen (once every N years in average), but may not yet have been observed. Climate warming brings other difficulties in their estimation: the definition of a return level has to be changed because a past and a future period are not prone to experience the same temperature levels, and the estimation is complicated. Different papers are devoted to this problem, for example: Cheng, L., AghaKouchak, A., Gilleland, E., & Katz, R. W. (2014). Non-stationary extreme value analysis in a changing climate. Climatic change, 127(2), 353-369 Du, T., Xiong, L., Xu, C. Y., Gippel, C. J., Guo, S., & Liu, P. (2015). Return period and risk analysis of nonstationary low-flow series under climate change. Journal of Hydrology, 527, 234-250 Obeysekera, J., & Salas, J. D. (2016). Frequency of Recurrent Extremes under Nonstationarity. Journal of Hydrologic Engineering, 21(5), 04016005 Parey, S., Malek, F., Laurent, C., Dacunha-Castelle, D. (2007). Trends and climate evolutions: statistical approach for very high temperatures in France. Clim.Change, 81, 331–352. Parey, S., Hoang, R.T.H., Dacunha-Castelle, D. (2010). Different ways to compute temperature return levels in the climate change context. Environmetrics, 21, 698–718.

Ans: Adaptation means preparation of baseline contingency plans for dealing with strong heat waves based on the current climatology. Such measures are not yet present in the territory and lead to many casualties each year. We wish to remark that the study domain is one of the hottest region in the world as mentioned in the

paper with the highest record-breaking temperature of 52°C in 2010. This region is a hub of agriculture activities and 50% of the population work outdoors. The local administrations have limited resources, therefore they want to prioritize the region for the adaptions like, early warning systems, introducing new the temperature tolerant crops, water management and providing shelters for the outdoors workers etc. Therefore, the information of return levels is good for the planning and adaptation strategies. So, a stationary analysis is already a pretty relevant contribution for the region. Moreover, we consider the stationary extreme value analysis due to short duration of the data (33 years) and to have reliable estimates with less uncertainty. Clearly, considering non-stationarity is a good idea for future work. We might consider using the centennial NCEP reanalysis [Compo et al., 2011], and looking at non-stationary extreme events using the methodology proposed in e.g. Parey et al. (2007) or Cheng et al. (2014).

References

Cheng, L., AghaKouchak, A., Gilleland, E. and Katz, R. W.: Non-stationary extreme value analysis in a changing climate, Clim. Change, 127(2), 353–369, doi:10.1007/s10584-014-1254-5, 2014.

Compo, G.P., J.S. Whitaker, P.D. Sardeshmukh, N. Matsui, R.J. Allan, X. Yin, B.E. Gleason, R.S. Vose, G. Rutledge, P. Bessemoulin, S. Brönnimann, M. Brunet, R.I. Crouthamel, A.N. Grant, P.Y. Groisman, P.D. Jones, M. Kruk, A.C. Kruger, G.J. Marshall, M. Maugeri, H.Y. Mok, Ø. Nordli, T.F. Ross, R.M. Trigo, X.L. Wang, S.D. Woodruff, and S.J. Worley, 2011: The Twentieth Century Reanalysis Project. Quarterly J. Roy. Meteorol. Soc., 137, 1-28. http://dx.doi.org/10.1002/qj.776

Cornes, R. C., and P. D. Jones, 2013: How well does the ERAInterim reanalysis replicate trends in extremes of surface temperature across Europe? J. Geophys. Res., 118, 10 262– 10 276, doi:10.1002/jgrd.50799.

Parey, S., Malek, F., Laurent, C., Dacunha-Castelle, D. (2007). Trends and climate evolutions: statistical approach for very high temperatures in France. Clim. Change,

81, 331.

---

## Referee Comment (RC4) · Anonymous Referee #3 · 20 Feb 2017

Return levels of temperature extremes in southern Pakistan by M. Zahid and co-authors

The study deals with the analysis of return levels of daily temperature extremes in the southern part of Pakistan, considering both daily maximum temperature and daily maximum wet-bulb temperature. The latter is important, as it affects the well-being of humans. The study used both observations and ERA-Interim re-analyses at the nearest grid points, both the original re-analyses and bias-corrected values. Given the overall increasing trend in the frequency or intensity of extremes related to daily maximum temperatures, such an analysis could led to very relevant results. This study, however, does not present such an analysis according to international standards and, therefore, in my view is not suitable for proceeding beyond the discussion part of Earth System Dynamics before a major revision of the manuscript. These are my main reservations:

The motivation: It does not become clear, why the re-analysis data are relevant here, except that they might be used to fill gaps with missing observations. In this respect, it appears important to correct the re-analyses for biases. That would, however, require a more advanced method as the one used here, combining local information at the stations with information on large-scale conditions.

The presentation of the methodology: The presentation of the methodology fills a rather large part of the manuscript, although much of it is widely used. Therefore, this part of the manuscript could be shortened.

The presentation of the results: The presentation of the results is not very concise. Numerous numbers and maps are included in the manuscript, but often they are not properly presented.

The discussion of the results: The results of the study are not really discussed, neither with respect to the scientific literature nor with respect to the underlying physical mechanisms and, only partly, with respect to the representativeness of the results for southern Pakistan for the rest of Pakistan or the rest of the wider region.

Concluding section: The concluding section is just a repetition of the main results of the study, and no conclusions of this study are given.

Tables: As for Table 2, it is not clear, why monthly mean values of the daily minimum and mean temperatures are presented here. As for Table 3, I am puzzled by the substantially different behaviour of the p-values according to the KS-test and the p-values according to the AD-test. As for Table 4, I am missing the units.

Figures: Generally, the figures and/or figure captions are lacking units. Also, in many cases the use of different plotting ranges for panels, which show the same kind of estimates for different data sets or different locations make it difficult to draw firm conclusions from these figures. As for Figs. 8 and 9, it layout of the panels makes is very hard to extract the relevant information from the map, since it the information on the

magnitude is hidden in the respective column including the symbols.

References: Most of the references to the scientific literature are used in the Introduction and in the methodology section, also highlighting the fact that the sections on the results and the discussion are not properly done.

In addition to these general concerns, I have numerous specific concerns, which might be relevant for the revised version of the manuscript.

———————————————

---

## Author Comment (AC4) · 27 Feb 2017

We would like to thank anonymous Referee # 3 for the critical review and comments. The step-by-step responses (*) to the main reservations of the referee are as follows.

Why we did not analyze the overall increasing trends in the frequency or intensity of extremes related to daily maximum temperatures?

(*)We did not analyze the increasing trends in the frequency and intensity of extremes related to daily maximum temperatures, and considers the stationary extreme value analysis due to short duration of the data (33 years) and to have reliable estimates with less uncertainty. Moreover, the study domain is one of the hottest region in the world as mentioned in the paper with the highest record-breaking temperature of 52°C in 2010. This region is a hub of agriculture activities and 50% of the population work outdoors.

[Figure]

The local administrations have limited resources, so they want to prioritize the region for the adaptions like, early warning systems, introducing new the temperature tolerant crops, water management and providing shelters for the outdoors workers etc. Therefore, the information of return levels is good for the planning and adaptation strategies. So, a stationary analysis is already a pretty relevant contribution for the region. Clearly, considering non-stationarity is a good idea for future work. We might consider using the centennial NCEP reanalysis (Compo et al., 2011).

\*\*\*\*\*\*\*\*\*\*\*\*\*\*\*\*\*\*\*\*\*\*\*\*\*\*\*\*\*\*\*\*\*\*\*\*\*\*\*\*\*\*\*\*\*\*\*\*\*\*\*\*\*\*\*\*\*\*\*\*

The motivation: It does not become clear, why the re-analysis data are relevant here, except that they might be used to fill gaps with missing observations. In this respect, it appears important to correct the re-analyses for biases. That would, however, require a more advanced method as the one used here, combining local information at the stations with information on large-scale conditions.

(\*) It is common practices among meteorologists to use ERA Interim (or NCEP) data to study the local to regional to large scale climatic properties. These datasets are also often used to assess the skill of climate models. Therefore, it seems reasonable to include them here. Additionally, the ERA Interim reanalysis data was proved to be very good at replicating trends in percentile-based measures of temperature (Cornes and Jones, 2013). However, it is still not clear that ERA data can simulate well meteorological extremes. This is why we use ERA Interim data to see how well it performs in the target area against observations. We are well aware – this is clearly explained in the text – that one could used more advanced bias correction methods. But here we want to show whether if we reduce to zero the bias in the first two moments (note that most scientists and practitioners focus only on these two statistical properties), we are still able to have a good representation of the tail of the distribution. In some stations like Nawabshah, Karachi etc, even the standard bias correction show very good agreement with observations. However, we agree with reviewer that if ERA data has to be used in the region (and elsewhere) to study extremes, a more advance method is needed. We

wish to underline the need to loon into actual station data. We have added this detail in the revised version of the paper.
* * *
The presentation of the methodology: The presentation of the methodology fills a rather large part of the manuscript, although much of it is widely used. Therefore, this part of the manuscript could be shortened. The presentation of the results: The presentation of the results is not very concise. Numerous numbers and maps are included in the manuscript, but often they are not properly presented.

(*) We agree that the statistical methods used here method is widely used, but, we prefer giving some details in order to address an audience that might be not so familiar with extreme value theory. We agree that the presentation includes a lot of maps and figures, but we remind the reviewer that, as mentioned in the paper, this is the first analysis for extremes using extreme value theory in Pakistan. Therefore, we consider giving all the possible details, to provide a thorough picture of the methodology and results to the fellow researches in Pakistan and neighboring countries.
* * *
The discussion of the results: The results of the study are not really discussed, neither with respect to the scientific literature nor with respect to the underlying physical mechanisms and, only partly, with respect to the representativeness of the results for southern Pakistan for the rest of Pakistan or the rest of the wider region.

(*) The results are discussed with respect to the available scientific literature. Please see the following cited references in results and discussion.

Sacrrott and MacDonald, 2012 (line 17, page 7), Coles, 2001 (line 19, page 7), Furrer et al., 2010 (line 20, page 7), Davison and Smith, 1990 (line 26, page 7), Hatfield and Preuger, 2015 (line 29, page 9).

The scientific literature regarding extreme value theory and return levels is not available

(mentioned between line 13 -14 page 3) for this specific region, therefore we could not discuss it in results. Understanding the meteorological mechanisms behind heat waves is well beyond the scope of this paper, which is only mostly statistical in nature. Much more work at this regard would be needed. Note that we have clearly explained why the statistical properties analyzed here are relevant for human welfare and economy in the region.
* * *
Concluding section: The concluding section is just a repetition of the main results of the study, and no conclusions of this study are given.

(*) The concluding section is named as "Summary and Conclusion", therefore we have summarized the results in the beginning and conclusions are given between lines 4-17 page 11. However, it is customary to summarize the results of the paper at the beginning of the last section, especially for a paper where statistical properties are analyzed.
* * *
Tables: As for Table 2, it is not clear, why monthly mean values of the daily minimum and mean temperatures are presented here. As for Table 3, I am puzzled by the substantially different behaviour of the p-values according to the KS-test and the p-values according to the AD-test. As for Table 4, I am missing the units.

(*) Table 2: shows the mean monthly climatic characteristics of the region from 1980-2010. It is there to describe the climatology of the region. This is a useful complement to the analysis of the extremes performed in the rest of the paper.

Table 3 : shows the different behavior of p-values because KS test and AD test are two different methods, and are used here to see the goodness-of-fit at each station. For details please see line 9-12 page 8.

Table 4 : show the estimated parameters shape $\xi$, scale $\sigma$ and standard error $\Delta\xi$ of all

the data sets.

Shape parameter $\xi$ has no unit. Scale parameter $\sigma$ has the unit "degree Celsius" like temperature.

\*\*\*\*\*\*\*\*\*\*\*\*\*\*\*\*\*\*\*\*\*\*\*\*\*\*\*\*\*\*\*\*\*\*\*\*\*\*\*\*\*\*\*\*\*\*\*\*\*\*\*\*\*\*\*\*\*\*\*

Figures: Generally, the figures and/or figure captions are lacking units. Also, in many cases the use of different plotting ranges for panels, which show the same kind of estimates for different data sets or different locations make it difficult to draw firm conclusions from these figures. As for Figs. 8 and 9, it layout of the panels makes is very hard to extract the relevant information from the map, since it the information on the magnitude is hidden in the respective column including the symbols.

(\*) The units are placed inside the figures, but now we have written them in figure captions as well. Regarding Figure 8 and 9 we think two different colors clearly distinguish between observations and bias corrected ERA Interim return levels, also different symbols are used to differentiate among the cities and return level values. In our point of view, the information on the magnitude of extremes is quite obvious here. However, suggestions to improve Figure 8 and 9 are welcome. Unfortunately it is never easy to find optimal solutions for that kind of figures. We have relied on interactions with colleagues and practitioners in multiple poster and oral presentations to gain inputs on that.

\*\*\*\*\*\*\*\*\*\*\*\*\*\*\*\*\*\*\*\*\*\*\*\*\*\*\*\*\*\*\*\*\*\*\*\*\*\*\*\*\*\*\*\*\*\*\*\*\*\*\*\*\*\*\*\*\*\*\*\*\*\*

References: Most of the references to the scientific literature are used in the Introduction and in the methodology section, also highlighting the fact that the sections on the results and the discussion are not properly done.

(\*) Given the nature of the paper (first analysis of extremes in the region), it seems quite natural that most of the referencing goes in the introduction and in the methodology. In results and discussion, following references are cited, highlighting the fact that it is

properly done.

Sacrrott and MacDonald, 2012 (line 17, page 7), Coles, 2001 (line 19, page 7), Furrer et al., 2010 (line 20, page 7), Davison and Smith, 1990 (line 26, page 7), Hatfield and Preuger, 2015 (line 29, page 9).

\*\*\*\*\*\*\*\*\*\*\*\*\*\*\*\*\*\*\*\*\*\*\*\*\*\*\*\*\*\*\*\*\*\*\*\*\*\*\*\*\*\*\*\*\*\*\*\*\*\*\*\*\*\*\*\*\*\*

References

Compo, G.P., J.S. Whitaker, P.D. Sardeshmukh, N. Matsui, R.J. Allan, X. Yin, B.E. Gleason, R.S. Vose, G. Rutledge, P. Bessemoulin, S. Brönnimann, M. Brunet, R.I. Crouthamel, A.N. Grant, P.Y. Groisman, P.D. Jones, M. Kruk, A.C. Kruger, G.J. Marshall, M. Maugeri, H.Y. Mok, Ø. Nordli, T.F. Ross, R.M. Trigo, X.L. Wang, S.D. Woodruff, and S.J. Worley, 2011: The Twentieth Century Reanalysis Project. Quarterly J. Roy. Meteorol. Soc., 137, 1-28. http://dx.doi.org/10.1002/qj.776

Cornes, R. C., and P. D. Jones, 2013: How well does the ERAInterim reanalysis replicate trends in extremes of surface temperature across Europe? J. Geophys. Res., 118, 10 262– 10 276, doi:10.1002/jgrd.50799.

---

## Author Response (AR2)

**Response to Report #1**

S. Parey (Referee #2)
sylvie.parey@edf.fr

The authors would like to thank again Sylvie Parey for the comments that helped us to improve our manuscript. Our responses to the comments are as follows.

1. p9 l39: "When looking at the wet-bulb temperature ... show some overlap with the those derived from station observations" "the" seems useless before "those" ?

Ans: "the" is removed on page10, lines 364-365.

2. p10 l33: "Although the ERA Interim ... magnitude of the extremes, but it provides" I did not expect a "but" here.

Ans: "But" is replaced with "still", page11, line 429.

3. p10 l38: "Although the bias corrected ERA Interim ... with the meteorological station data, statistically différences remain" something seems to be missing between "statistically" and "differences" (like significant for example)

Ans: word "yet" is added to complete the sentence, page12, line 442

4. Table 4 legend: "and standard error Δξ" I would have mentioned both "and standard errors Δξ and Δσ" as both are in the table

Ans: Δσ is added in the caption of table 3, page18.

**Response to Report #2**

Pingping Luo (Referee# 4)
robertluoping@126.com

The authors would like to thank Pingping Luo for the constructive comments and suggestions that helped us to improve our manuscript. Our point-by-point responses to the comments are given below.

1. Please use the continuous line number.

Ans: The continuous line number is provided.

2. Introduction section, I think it need restructure the order and paragraph in this section. Some small paragraph should connected with another paragraph.

Ans: The introduction section is modified.

3. Page3-4, from Page 3 Line 22 to Page 4, line 5, It need make it shortly. It is difficult to identify your objectives.

Ans: The objectives are modified and revised, page 3, between lines 92-110.

4. Table 3, The first part of "Observed Tmax", the format of line should be same with the others.

Ans: It has been formatted. Please see page 17.

5. Figure 4, the figure is not clear because there are some background in the figure. Please try to delete the background and make the figure clearly.

Ans: Background is removed.

6. Figure 5, the figure is not clear because there are some background in the figure. Please try to delete the background and make the figure clearly.

Ans: Background is removed.

7. I suggest the author should add some discussion on how your work connected with the extreme storm or dry. What your result can be contribute for? Such as flood management and so on. What is your research limitation.

Ans: Thank you for the suggestion. In this manuscript we focus on high temperature extremes, but we are well aware that other extremes of potentially huge impact exist in the region. Your reference to flood risk is most relevant, indeed. We have already prepared another manuscript on the precipitation extremes for flood management authorities, where we have linked the temperature and precipitation extremes. Besides temperature, an increase in transport of moisture flux from the Arabian Sea is also considered responsible for the intense precipitation in southern Pakistan (Kalim and Shouting, 2012, Freychet et al. 2015). We have added few lines 384-385 and 417 - on page 10 & 11 to give our readers an idea about the impacts of temperature extremes.

The current paper is only focusing on two types of temperature extremes due to their direct impact on crops, livestocks and society. The main aim of this publication is to guide the local administrations for preparation of baseline contingency plans to deal with strong heat waves based on the current climatology. Such measures are not yet present in the territory and lead to many casualties each year. This is already mentioned in the paper between lines 453 – 454 on page 12.

The sparse network of weather stations and lack of long-term data are the main limitations of this study. This is more explicitly mentioned in the manuscript now.

Our results are informative of the spatial patterns of extremes. The results and maps of this paper will be displayed in a freely available web-tool SindheX developed for the policy makers [www.sindhex.org]. This tool will be available online after the publication of our results. Our results will not only contributes to the regional planning, but can also be useful for the ongoing EU projects (SUCCESS, CSCCC), World Bank project (Sindh Resilience Project) and mega construction projects like China-Pakistan Economic Corridor (CPEC).

Clearly, the sparse network of weather stations and lack of long-term data are the main limitations of this study.

We agree with the reviewer's statement that in general "Just because it has not been done before is no justification for doing it now". But we honestly do not see why this applies to this study. This would be the case if a) the region we are investigating is of no interest, and this seems blatantly untrue; or b) the method we are using is mathematically ill-defined or of no interest; this also seem unlikely.

Figure 4 (page 10) of Rasul et al. (2012), is showing a time series of area weighted annual averages for mean daily temperatures in Pakistan. This figure includes annual temperature values of entire Pakistan, not only Sindh (our study area) from 1960 to 2010. Figure 11 (page 19), shows average frequency of heat waves over entire Sindh under limited conditions (at ≥ 40°C & ≥ 45°C for 5,6,7 consecutive days) from 1960 to 2009. Both figures are showing average trends with different durations and cannot be compared with this paper results. One must not assume that there are significant trends in the data just on the basis of these figures.

We remark that temperature extremes vary from one station to another in this region. The policy makers do not agree to implement adaptations just on the basis of analysis of extremes on spatially averaged temperature data, while it is important to study the stations individually for supporting more efficiently the definition of risk assessment.

Ans:. We gave detailed results in order to address an audience that might be not so familiar with this method. It first analysis for extremes using POT in Pakistan, so that it is clear that some of our potential readers might be not so familiar with this method. Note that many studies on climate extremes (including what reported in the IPCC reports) do NOT use extreme value theory (in the 2012 special IPCC report on extremes, EVT is vaguely mentioned and basically no results based on it are reported there). Therefore, we prefer to give all the possible details.

The shape parameters of all datasets are negative at all stations (Fig 4). This is described on page 8, lines 297-308. The spatial pattern of scale parameter in all datasets is added on page 8 between lines 310-315.

Figure 8 and 9 are made for the detailed spatial overview of the temperature extremes in Sindh so that all stakeholders can see the return values in each station. We have added more explanation of Figures (8 and 9) on page 10, lines 371 - 394.

Table containing the mean climatic characteristics of Sindh is given as a supplement material.

4. The "Summary and conclusion" section is too long. The authors should concentrate on the main results and conclusions of the current research. The introduction section is also too long and should be shortened. E.g. the sixth paragraph of "Summary and conclusion" should be deleted, which is suitable for the introduction part.

What's more, the main conclusions are too specific, concentrating on some detailed data/facts. Earth System Dynamics is an international journal, the results and conclusion part should report more meaningful findings to the scientific community.

Ans: The "Summary and conclusion" and "Introduction" part have been revised.

This is a maiden study of probability of occurrence of extremes in the region using extreme value theory. It seems to us that it makes sense to report them in detail. The findings are meaningful not only for the region but neighboring countries like India, Iran, and  for many ongoing EU, WB and Chinese projects (SUCCESS, CSCCC, Sindh Resilience Project, CPEC) in Sindh. Clearly, there is much scope for understanding the dynamical processes behind such extremes, but this is well beyond of the goal of the present paper.

5. The bias correction method. The bias correction formula introduces the information of observed data which will surely improve the results of ERA data. On the one hand, we don't need the ERA data when observed data are available for the past, so the bias correction method proposed by the authors seems to be not necessary; on the other hand, we need ERA or other reanalysis data to project future climate extremes. However, this method needs the observed data to make the correction. What's the usage of this bias correction method?

As mentioned earlier the observations network is too sparse in the region and it is very difficult to have long-term reliable data. We could hardly achieve appropriate observational data of nine weather stations for this study. We tested ERA Interim to use it as an alternative for the regions within Sindh lacking data.

Our analysis wishes to show two things 1) ERA Interim is "as it is" not a good enough datasets to look at extremes in this region. The statistics of extremes is completely wrong. Note that reanalysis datasets are often used to construct climatologies (and to study climate change). 2) If we correct the first two moments of the ERA-Interim climatology so that it matches the observed ones (this amounts to correcting the bulk statistics, or the vast majority of the events), in many cases we are still unable to describe well enough the statistics of extremes. Bias correction methods tailored at extremes are needed, instead, if one wants to use ERA-Interim data to look meaningfully at extremes.

**List of all changes in the manuscript**

**Changes according to Reviewer # 2**

- Lines 364-365, p10, "the" is removed.
- Line 429, p11, "But" is replaced with "still".
- Line 442, p12, word "yet" is added to complete the sentence.
- $\Delta\sigma$ is added in the caption of table 3, p18.

**Changes according to Reviewer # 4**

- The continuous line number is provided.
- The introduction section is modified.
- Lines 92- 110, p3, the objectives are modified and revised.
- It has been formatted, p17.
- Background is removed, p20.
- Background is removed, p21.
- Lines 384-385 and 417 – 419 added on p10 & p11.
- Two suggested references by the reviewer are added on p10 &11.

**Changes according to Reviewer #5**

- Lines 151-158, p4, trend detection is explained. Mann Kendall trend test is also included for testing the presence of trends.
- Lines 297-308 & 310-315, p8, the spatial patterns of the shape and scale parameters in all datasets are explained.
- Lines 371-394, p10, more explanation of Figures (8 and 9) are added.
- The "Summary and conclusion" and "Introduction" part have been revised.

[revised manuscript text omitted]

---

## Author Response (AR3)

**Response to Report #1**

Anonymous Referee

The authors would like to thank Anonymous Referee for giving valuable comments that have improved the quality of this manuscript. Our responses to the comments are given below.

1. Please modify the English grammar again.

Ans: Done.

2. The row spacing of the contents of the table 1 should be consistent.

Ans: The row spacing in Table 1 is formatted.

3. I think the format of table 2 should be adjusted appropriately in order to better present the reader.

Ans: Table 2 is adjusted.

4. Page 19, line769-771. The number covers the font. Please adjust its format.

Ans: Done.

5. Page 22, line844-846. The number covers the font. Please adjust its format.

Ans: Done.

6. Page 23, line859-860. The number covers the font. Please adjust its format.

Ans: Done.

7. Page 24, line871-873. The number covers the font. Please adjust its format.

Ans: Done.

8. Page 25, line885-886. The number covers the font. Please adjust its format.

Ans: Done.

8. Figure 5, Please make this figure more clearly. The Horizontal coordinate and Vertical coordinate should be black color line. Please check it with other figures.

Ans: The black color line is added to horizontal and vertical coordinates in Figure 5.

**Response to Report #2**

S. Parey (Referee #2)
sylvie.parey@edf.fr

The authors would like to thank Sylvie Parey for the keen comments and suggestions that have substantially improved our manuscript. The point by point reply to the comments are given below.

1. p4 line 140: "statistical effects associated to the presence discrete values" I would have expected "to the presence of discrete values"

Ans:   Done. See page4, line 146.

2. p11 line 422: "the most vulnerable people are those who are involve": involved

Ans:   Done. See page12, line 444.

3. p11 line 427: "We found that the RLs of stations and ERA Interim showed differences are between ..." are does not seem useful here.

Ans:   Done. See page12, line 449.

4. p11 line 442: "yet statistically differences remain " do you mean statistically significant differences?

Ans:   Yes, we have added "significant". See page 12, line 465.

5. p11 line 452: "Our results will not only contributes" the final "s" is not needed.

Ans:   Done.

[revised manuscript text omitted]